# CODESYNC: Synchronizing Large Language Models with Dynamic Code Evolution at Scale

**Chenlong Wang** [* 1]  **Zhaoyang Chu** [* 1]  **Zhengxiang Cheng** [* 1]  **Xuyi Yang** [2]  **Kaiyue Qiu** [1]  **Yao Wan** [1]
**Zhou Zhao** [3]  **Xuanhua Shi** [1]  **Hai Jin** [1]  **Dongping Chen** [1 †]

## Abstract

Large Language Models (LLMs) have exhibited exceptional performance in software engineering yet face challenges in adapting to continually evolving code knowledge, particularly the frequent updates of third-party library APIs. This limitation, rooted in the static pre-training datasets, often results in non-executable code or implementations with suboptimal safety and efficiency. To this end, we introduce CODESYNC, a data engine to identify outdated code patterns and collect real-time code knowledge updates from Python third-party libraries at scale. Building upon CODESYNC, we develop CODESYNCBENCH, a comprehensive benchmark for assessing LLMs' ability to stay *synchronized* with code evolution, which covers real-world updates for 220 APIs from six Python libraries. Our benchmark offers 3,300 test cases spanning three evaluation tasks and an update-aware instruction tuning dataset of 2,200 training samples. Extensive experiments on 14 LLMs reveal that they struggle with dynamic code evolution, even with the support of advanced knowledge updating methods (*e.g.*, DPO, ORPO, and SimPO). Our CODESYNC lays a strong foundation for developing more effective and robust methods for real-time and large-scale code knowledge updating in the future. The experimental code is available at: https://github.com/CGCL-codes/naturalcc/tree/main/examples/codesync.

---

[*]Equal contribution [1]National Engineering Research Center for Big Data Technology and Systems, Services Computing Technology and System Lab, Cluster and Grid Computing Lab, School of Computer Science and Technology, Huazhong University of Science and Technology, Wuhan, China [2]Wuhan University [3]Zhejiang University. Correspondence to: Yao Wan <wanyao@hust.edu.cn>.

*Proceedings of the 42nd International Conference on Machine Learning*, Vancouver, Canada. PMLR 267, 2025. Copyright 2025 by the author(s).

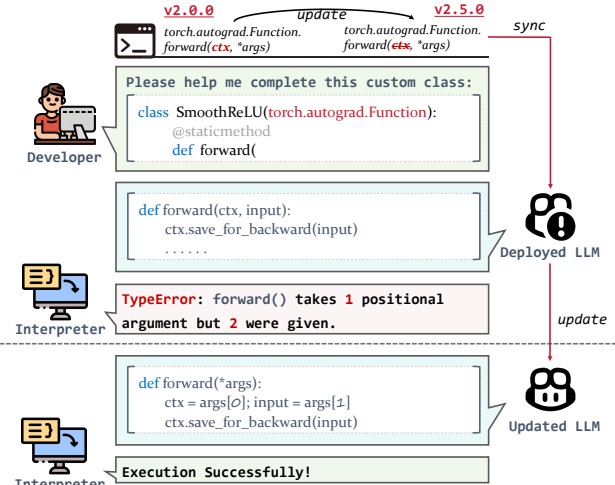

Figure 1: **LLMs struggle to adapt to API updates, leading to potential compatibility issues in generated code.** For example, the `device` parameter was removed from the `full` function in `numpy` version 2.1.0, making LLM failed to provide correct invocation. It highlights the need for API knowledge updating to synchronize LLM with the latest API changes and correctly generate updated API invocations.

## 1. Introduction

Large Language Models (LLMs), exemplified by DeepSeek-R1 (Guo et al., 2025), CodeLlama (Roziere et al., 2023), and GPT-4o (OpenAI, 2024), have demonstrated remarkable performance in automating software development through generating executable code (Jiang et al., 2024). However, due to static pre-training datasets, they often struggle to adapt to the rapidly evolving knowledge in programming, especially the frequent updates of external library APIs (Tao et al., 2012; Zhang et al., 2020).

As illustrated in Figure 1, when prompted to create an array on a CUDA device, the LLM is unaware of the removal of the `device` parameter in the updated `numpy.full` function. This oversight results in an error, *i.e.*, *"TypeError: full() got an unexpected keyword argument 'device'"*. The pitfalls of generating code containing outdated APIs can lead to parameter compatibility issues, which cause

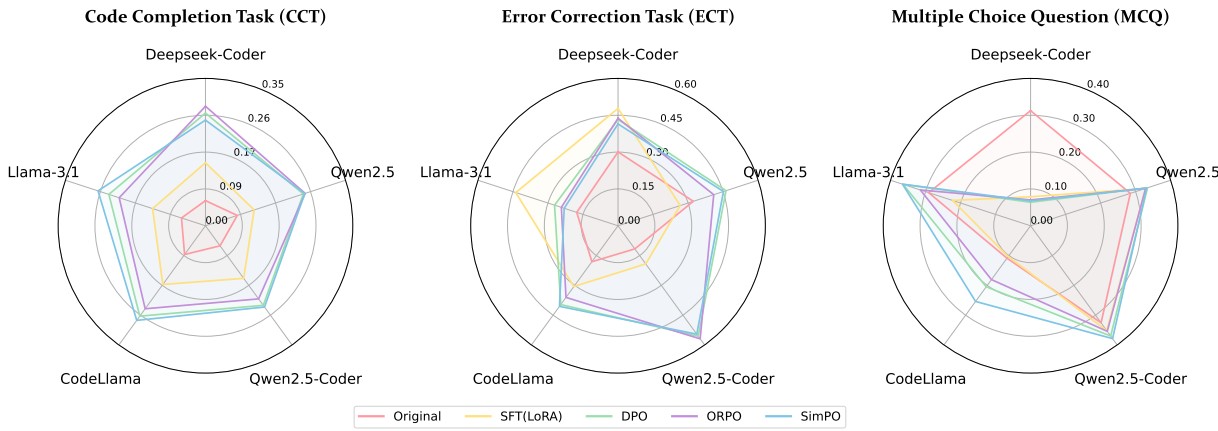

Figure 2: **Performance comparison of knowledge updating methods across three evaluation tasks on five LLMs.** All LLMs shown in the figure are instruction-tuned versions. The results reveal that LLMs face challenges in adapting to dynamic API updates, even with the support of knowledge updating approaches, emphasizing the need for improvements in real-time code knowledge updating.

programs to crash or malfunction, undermining the stability and reliability of software (Bai et al., 2024; Zhang et al., 2024c). This challenge highlights the need for LLMs to *synchronize* with the dynamic evolution of practical code knowledge, particularly the fast-paced API updates that have immediate and visible impacts on software development.

Recently, Liu et al. (2024c) made an initial attempt to address this gap by benchmarking LLMs' ability to access API updates through fine-tuning. However, their benchmark relies on **unauthentic** API updates synthesized by GPT-4 (OpenAI, 2024) rather than real-world library updates, resulting in potentially biased assessments of LLMs' adaptability to practical code evolution. We argue that an authentic evaluation system should be established to answer the key question: *Can LLMs be effectively and efficiently updated to handle real-time API modifications?*

To address this gap, this paper introduces CODESYNC, a scalable data engine for collecting authentic code knowledge updates from Python third-party libraries across various domains, including data science (*e.g.*, pandas), artificial intelligence (*e.g.*, torch), and web development (e.g., flask). Specifically, CODESYNC systematically identifies real-time API updates at scale by tracking changes to API signatures across library versions. For each identified API with updates, it retrieves relevant code instances invoking the API from GitHub repositories using GitHub Code Search (GitHub). Based on these real-world API invocations, CODESYNC employs DeepSeek-V3 (Liu et al., 2024a) to synthesize contrastive invocations for the legacy and updated API versions.

Based on CODESYNC, we develop CODESYNCBENCH, an extensive benchmark for assessing LLMs' ability to stay *synchronized* with dynamic code evolution, which includes real-world updates for 220 APIs (130 functions, 59 initializers, and 31 methods) from 6 Python libraries, along with 3,300 legacy-updated pairs of API invocation instances. The benchmark provides 3,300 test cases across three evaluation tasks, *i.e.*, *Code Complete Task* (CCT), *Error Correction Task* (ECT), and *Multiple Choice Question* (MCQ), accompanied by an update-aware instruction tuning dataset comprising 2,200 training samples. Unlike retrieval-augmented frameworks that enhance LLMs at the expense of increased inference overhead and without reflecting true model updates, CODESYNCBENCH focuses on evaluating and improving LLMs' ability to internalize API update knowledge and accurately recall it during code generation.

**Take-Aways.** We benchmark 14 state-of-the-art LLMs (*e.g.*, ChatGPT (OpenAI, 2024), DeepSeek (Liu et al., 2024a) and Claude (Anthropic, 2024)), including both proprietary and open-source models, as well as five knowledge updating methods (*e.g.*, DPO (Rafailov et al., 2023), ORPO (Hong et al., 2024), and SimPO (Meng et al., 2024)). Our findings reveal several key insights. First, as shown in Figure 2, assessment results indicate that LLMs struggle to adapt to dynamic API updates, even with the support of advanced knowledge updating approaches, highlighting the need for further advancements in real-time code knowledge updating. Moreover, the number of API invocations available for training and the types of updated APIs significantly impact the effectiveness of knowledge updating, increasing the complexity of handling real-world API modifications.

**Contributions.** Our primary contributions are summarized as follows.

- **A Data Engine.** We introduce CODESYNC, a data engine that systematically collects real-time code knowledge updates from various Python third-party libraries.

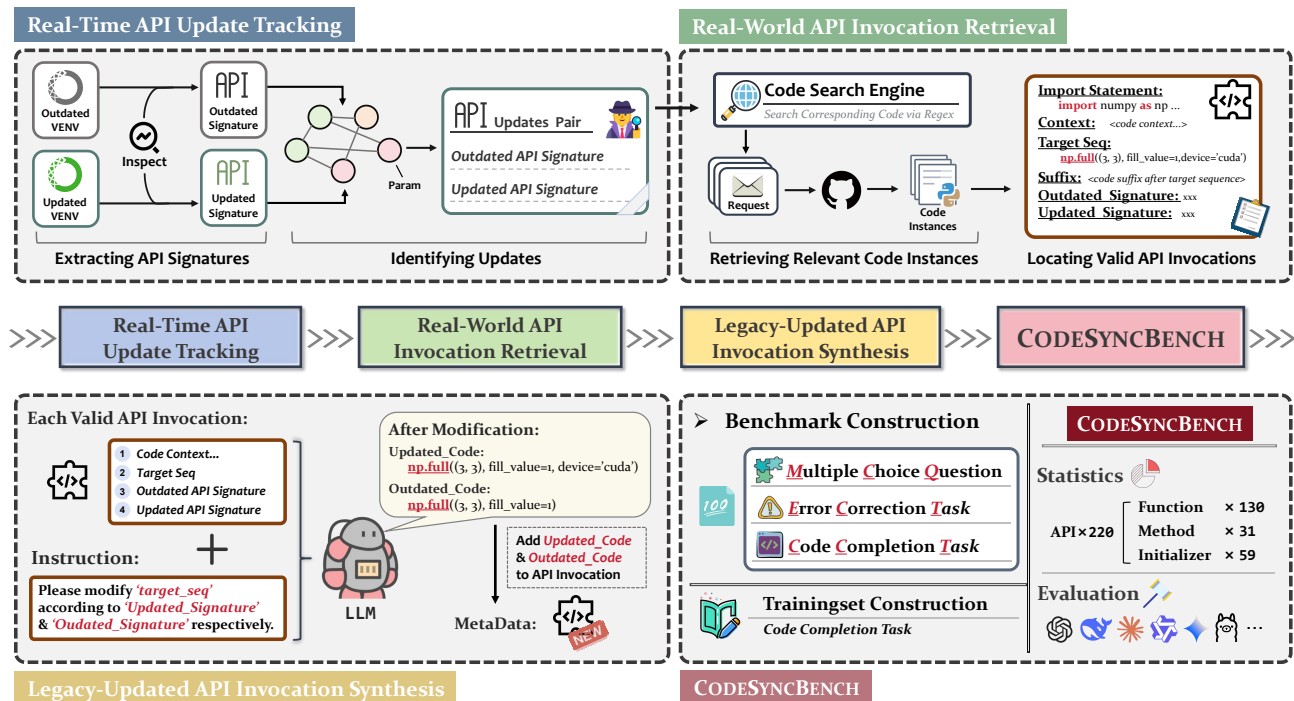

Figure 3: **An overview of our proposed CODESYNC framework.** CODESYNC consists of four key steps: **(1) Real-Time API Update Tracking** tracks and collects API updates by comparing legacy and latest versions of libraries. **(2) Real-World API Invocation Retrieval** is designed to crawl API invocations and locate valid API calls. **(3) Legacy-Updated API Invocation Synthesis** leverages LLMs to synthesize new API invocation statements based on legacy and updated signatures, respectively, and then recognizes them into metadata. **(4) CODESYNCBENCH** is used to evaluate the performance of LLMs on API updating tasks, with a period spanning from January 1, 2023 (post-GPT-3.5 release) to current versions.

- **A Novel Benchmark.** We develop CODESYNCBENCH, a novel benchmark covering updates for 220 APIs across six Python libraries. It offers 3,300 test cases across three evaluation tasks and an update-aware instruction tuning dataset with 2,200 training samples. This benchmark can serve as a rigorous testbed to facilitate the development of real-time code knowledge updating methods.

- **Comprehensive Evaluation.** Our extensive experiments on 14 state-of-the-art LLMs, including both proprietary and open-source models, indicate that they still struggle to handle dynamic code evolution. Additionally, our results reveal that knowledge updating methods can improve LLM synchronization with API updates, though challenges remain to be addressed.

## 2. CODESYNC: A Data Engine for Real-Time Code Knowledge Collection

As illustrated in Figure 3, we propose CODESYNC, a data engine for real-time collection of code knowledge evolution, which operates through three key steps: **(1) Real-Time API Update Tracking.** CODESYNC identifies and extracts API updates across diverse Python third-party libraries by

systematically tracking changes to API signatures between library versions (see Section 2.1). **(2) Real-World API Invocation Retrieval.** For each identified API with updates, CODESYNC retrieves relevant code instances invoking the API from GitHub repositories through GitHub Code Search (GitHub) (see Section 2.2). **(3) Legacy-Updated API Invocation Synthesis.** Building on the retrieved real-world API invocations, CODESYNC employs DeepSeek-V3 (Liu et al., 2024a) to synthesize contrastive code instances that invoke legacy and updated APIs, respectively (see Section 2.3). Based on CODESYNC, we establish CODESYNCBENCH, a benchmark for assessing real-time code knowledge of LLMs, which collects updates for 220 APIs (including 130 functions, 59 initializers, and 31 methods) from 6 Python libraries, totaling 3,300 legacy-updated pairs of API invocation instances (see Section 2.4).

### 2.1. Step 1: Real-Time API Update Tracking

The functionality of APIs is exposed through their signatures, which provide an interface for developers to utilize this functionality within code. This feature enables systematic tracking of library API updates by monitoring changes in their signatures.

Table 1: **Statistics of tracked API updates.** We systematically identify API updates across diverse Python third-party libraries by monitoring changes in API signatures between the latest version and an outdated version around January 1, 2023. This period coincides with the introduction of the milestone GPT-3.5.

| Library | Legacy Version | Updated Version | Num. |
|---|---|---|---|
| pandas | 2.0.3 | 2.1.4 | 1,043 |
| numpy | 1.24 | 2.1 | 55 |
| scipy | 1.10.0 | 1.13.1 | 494 |
| tensorflow | 2.11.0 | 2.18.0 | 161 |
| torch | 2.0.0 | 2.5.0 | 4,260 |
| flask | 2.2.2 | 3.0.0 | 22 |

**Extracting API Signatures.** We target 6 widely used Python third-party libraries: pandas, numpy, scipy, tensorflow, torch and flask. To collect complete API signatures from these libraries, we leverage Python's built-in *inspect* module, a *dynamic* reflection tool provided by the Python standard library (Python, b). This tool enables runtime analysis and collection of information about Python objects, including modules, classes, functions, and methods. For each library, we extract API signatures using *inspections* within virtual environments configured with specific library versions. Further details are provided in Appendix B.1.1.

**Identifying API Updates.** To evaluate LLMs' ability to synchronize with real-time API evolution, we consider the most recent library version before ChatGPT's release (OpenAI, 2023) as the legacy version and the current library version as the updated version. Then, we identify API updates by systematically comparing API signatures between versions. To determine whether an update exists for a given API, we perform a *static* analysis to establish parameter mappings for same-name APIs across versions. These mappings allow us to analyze API changes at the parameter level by examining differences in attributes such as parameter name, position, and type. Using this approach, we identify 6,063 API updates from the six targeted Python libraries, as summarized in Table 1. More implementation details are provided in Appendix B.1.2.

### 2.2. Step 2: Real-World API Invocation Retrieval

While API updates are reflected in signature changes, collecting this information alone is insufficient to fully capture the evolution of code knowledge. To address this, we consider real-world API invocation scenarios, focusing on modifications in API usage within actual code contexts. For each API update identified in Section 2.1, we collect relevant code instances that invoke the API from GitHub.

**Retrieving Relevant Code Instances.** We use GitHub Code Search (GitHub) to retrieve Python files that potentially contain API invocations by designing multiple matching templates. For example, to retrieve code invoking the function torch.nn.Linear, we match the API name (*e.g.*, .Linear) along with relevant import statements (*e.g.*, import torch.nn as nn and from torch import nn). Further details on the matching templates are provided in Appendix B.1.3.

**Locating Valid API Invocations.** Code instances retrieved via matching templates may only potentially invoke the target APIs, requiring precise localization to confirm valid invocations. To achieve this, we parse each code instance into an *Abstract Syntax Tree* (AST) using Python's built-in *ast* module (Python, a) and traverse all statements to identify those that genuinely contain targeted invocations. Moreover, we perform alias resolution on import statements to establish mappings between full module names (*e.g.*, numpy) and their aliases (*e.g.*, np), ensuring more accurate identification of valid API invocations. For example, we locate statements that contain np.full for the full function and nn.Linear for the Linear class initializer. Furthermore, regarding method invocation locating, the *ast* module enables us to track objects whose types match the target class by examining class instantiations and assignments. For example, in the case of x.reshape(), we identify that x is of type torch.Tensor, confirming a valid invocation of the reshape() method from the torch.Tensor class. The strategy guarantees that the filtered instances are absolutely correct. Detailed implementation is provided in Appendix B.1.4.

Through retrieval and localization, we filter out APIs with fewer than 15 valid invocation instances. Out of 6,036 APIs, 220 meet the criteria, each with 15 valid invocation instances, resulting in a total of 3,300 instances.

### 2.3. Step 3: Legacy-Updated API Invocation Synthesis

While real-world code instances with valid API invocations can be retrieved from GitHub repositories, it is challenging to determine the exact library version of the invoked API. To address this, we synthesize the contrastive API invocation pairs—legacy and updated—using state-of-the-art LLMs, which have demonstrated strong capabilities in revising code while preserving both semantic and syntactic correctness (Guo et al., 2024b).

Specifically, for each API invocation instance retrieved in Section 2.2, we prompt DeepSeek-V3 (Liu et al., 2024a) to adapt the target API invocation statement according to the legacy and updated API signatures, respectively, while preserving the integrity of the surrounding context. To ensure data quality, the authors manually verify the divergence between legacy and updated versions, instructing the LLM to re-synthesize cases with insufficient divergence. This approach ensures divergence in API usage while maintaining functional equivalence between legacy and

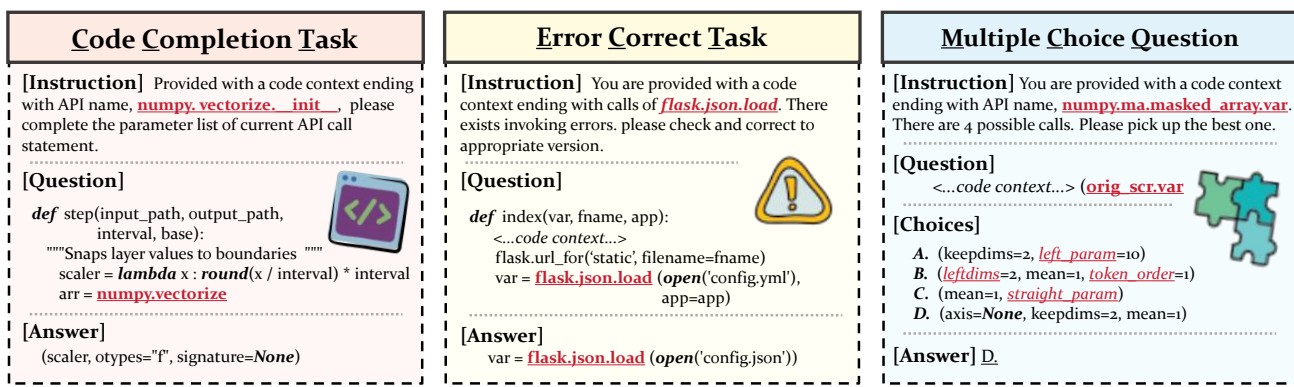

Figure 4: **An illustrative example of three evaluation tasks of CODESYNCBENCH. (1) CCT** only provides the API call name at the end of the question, without explicitly listing the parameters, expecting the completion. **(2) ECT** includes an incorrect parameter list at the end of the question, expecting the correction. **(3) MCQ** does not explicitly listing the parameters, but presents one correct option and three incorrect options, expecting the most accurate answer.

Table 2: **Statistics of data in CODESYNC.** We construct CODESYNCBENCH and the associated training set step by step, from identifying real-time API updates, retrieving real-world invocations, and synthesizing legacy-updated invocations to building training and test samples.

| Step | Setting | Input | Num. | Output | Num. |
|---|---|---|---|---|---|
| 1 | - | Python Libraries | 6 | API Updates | 6,036 |
| 2 | - | API Updates | 220 | API Invocations | 3,300 |
| 3 | - | API Invocations | 3,300 | Legacy-Updated Invocation Pairs | 3,300 |
| CODESYNC BENCH | Train | Legacy-Updated Invocation Pairs | 2,200 | Update-Aware Instructions | 2,200 |
| | Test | Legacy-Updated Invocation Pairs | 1,100 | CCT Tests | 1,100 |
| | | | | ECT Tests | 1,100 |
| | | | | MCQ Tests | 1,100 |

updated implementations, enabling explicit modeling of API evolution. Through this process, we synthesize 3,300 legacy-updated API invocation pairs from 3,300 real-world code instances. The detailed prompt is provided in Appendix C.1.

## 2.4. CODESYNCBENCH: A Benchmark for Real-Time Code Knowledge Assessment

Based on CODESYNC, we develop CODESYNCBENCH, a real-time benchmark for assessing how effectively LLMs adapt to evolving code knowledge, which comprises three evaluation tasks, including *Code Completion Task* (CCT), *Error Correction Task* (ECT), and *Multiple Choice Question* (MCQ), as shown in Figure 4. CODESYNCBENCH covers updates for 220 APIs across 6 Python libraries, including 130 functions, 59 initializers, and 31 methods. Each API is associated with 15 legacy-updated invocation pairs (3,300

in total), with 5 pairs for evaluation (1,100 in total) and 10 for training (2,200 in total). Based on this, our benchmark builds 1,100 tests per evaluation task, accompanied by a training set comprising 2,200 update-aware instructions, providing a rigorous foundation for assessing LLMs' ability to stay synchronized with API evolution.

**Code Completion Task (CCT) (Lu et al., 2021).** This task evaluates whether LLMs have internalized the updated APIs and can recall them during code generation. Given a code snippet ending with an API name, the LLM is prompted to complete the parameter list, with the updated API invocation statement serving as the ground truth. To measure the API invocation completion, we employ three widely used metrics: BLEU (Papineni et al., 2002) for evaluating lexical precision, ROUGE-L (Lin, 2004) for measuring semantic coverage, Relative Edit Distance (Ristad & Yianilos, 1998) for quantifying structural deviation, and CodeBLEU (Ren et al.) for assessing AST matching.

**Error Correction Task (ECT) (Zheng et al., 2024a).** This task simulates real-world *debugging* scenarios, where an interpreter throws an exception related to a specific API invocation. It evaluates the LLM's ability to actively correct potential errors. Given a code snippet ending with a legacy API invocation, the LLM is prompted to rectify it to the updated version. We assess the accuracy of API invocation correction using BLEU (Papineni et al., 2002), ROUGE-L (Lin, 2004), Relative Edit Distance (Ristad & Yianilos, 1998), and CodeBLEU (Ren et al.).

**Multiple Choice Question (MCQ) (Nguyen et al., 2025).** This task evaluates the LLM's ability to discriminate between correct and incorrect API invocations, requiring a deep internalization of the updated APIs. Given four candidate API invocations, including one correct answer and three plausible distractors, the LLM is prompted to select the optimal choice. The distractors, synthesized

Table 3: **The performance of different LLMs in accessing API updates.** We evaluate nine popular LLMs on CODESYNCBENCH, revealing their poor performance in API invocation tasks. The results highlight significant limitations in LLMs' ability to handle updated APIs, with even state-of-the-art models struggling to achieve high scores due to outdated knowledge. (BU for BLEU, RL for ROUGE-L, and RED for Relative Edit Distance)

| LLM | Knowledge Cutoff Date | CCT | | | ECT | | | MCQ | | |
|---|---|---|---|---|---|---|---|---|---|---|
| | | BU↑ | RL↑ | RED↓ | BU↑ | RL↑ | RED↓ | P@1↑ | P@3↑ | P@5↑ |
| *Closed Source Models* | | | | | | | | | | |
| GPT-4o | Oct. 2023 | 14.93 | 47.07 | 58.87 | 37.07 | **67.13** | **43.06** | **38.98** | **42.09** | **46.07** |
| GPT-4o-mini | Oct. 2023 | 7.45 | 32.39 | 67.14 | 33.69 | 51.06 | 49.54 | 29.58 | 34.63 | 35.58 |
| Claude-3.5-Sonnet | Apr. 2024 | **19.29** | 49.24 | **57.07** | **37.91** | 65.85 | 43.21 | 36.08 | 40.13 | 41.80 |
| Gemini-1.5-Pro | Nov. 2023 | 17.62 | **49.65** | 57.85 | 32.75 | 61.93 | 48.03 | 34.40 | 40.55 | 43.16 |
| *Open Source Models* | | | | | | | | | | |
| DeepSeek-V3 | Jul. 2024 | 19.24 | **44.13** | 57.67 | 51.57 | 62.64 | 34.12 | 31.54 | 34.41 | 35.78 |
| DeepSeek-R1 | Jul. 2024 | **19.32** | 44.09 | **57.54** | **51.81** | **62.76** | **34.05** | 31.61 | 34.41 | 35.78 |
| Qwen2.5-14B-Instruct | Mar. 2024 | 10.46 | 36.94 | 63.89 | 30.82 | 49.60 | 54.45 | **37.28** | **38.88** | **39.45** |
| Qwen2.5-32B-Instruct | Mar. 2024 | 13.97 | 39.43 | 62.24 | 40.31 | 55.58 | 42.81 | 35.35 | 37.50 | 38.16 |
| Qwen2.5-72B-Instruct | Mar. 2024 | 16.06 | 41.53 | 59.76 | 45.03 | 57.92 | 38.23 | 33.49 | 36.41 | 37.41 |

by DeepSeek-V3 (Liu et al., 2024a), include perturbations such as adding an invalid parameter, removing a required parameter, and rearranging parameter order. We employ the Pass@$k$ metric (Chen et al., 2021a) to measure the probability that the LLM passes a test case within $k$ attempts, which is calculated by drawing $n \geq k$ answers from the LLM for each test case and counting the number of correct answers $c \leq n$. We use $n = 10$ and $k \in \{1, 3, 5\}$ (abbreviated as P@1, P@3, and P@5).

**Training Set.** To evaluate knowledge updating methods, we build an instruction tuning dataset $\mathcal{D} = \{(\mathbf{i}, \mathbf{o}_{\text{old}}, \mathbf{o}_{\text{new}})\}$. As illustrated in Section E.1, $\mathbf{i}$ denotes an update-aware instruction containing a code snippet with an incomplete API invocation (*e.g.*, "`array=numpy.full(`"). $\mathbf{o}_{\text{old}}$ and $\mathbf{o}_{\text{new}}$ are output statements that accomplish the code. $\mathbf{o}_{\text{new}}$ represents the correct invocation with the updated API, while $\mathbf{o}_{\text{old}}$ reflects the legacy version. $\mathbf{o}_{\text{old}}$ and $\mathbf{o}_{\text{new}}$ share the same basic functionality, differing only in the parameters affected by the API update. The paired invocations allow the LLMs to identify update-related changes by computing token-level differences between $\mathbf{o}_{\text{old}}$ and $\mathbf{o}_{\text{new}}$.

## 3. Can LLMs *Sync* with Code Evolution?

To assess LLMs' ability to synchronize with code evolution, we investigate the following *Research Questions* (RQs):

- **RQ1: Benchmarking Large Language Models.** *Can LLMs access real-time API updates without relying on retrieval-augmented frameworks?*

- **RQ2: Benchmarking Knowledge Updating Methods.** *Can LLMs be effectively and efficiently updated to synchronize with API changes using knowledge updating methods without compromising model utility?*

- **RQ3: Impact of API Update Settings.** *How do different*

*API update settings,* e.g.*, the numbers of API invocations available for training and the types of updated APIs, impact the performance of knowledge updating?*

### 3.1. RQ1: Benchmarking Large Language Models

We benchmark nine state-of-the-art LLMs in accessing real-time API updates without retrieval-augmented settings, including four proprietary models (*i.e.*, GPT-4o, GPT-4o-mini (OpenAI, 2024), Claude-3.5-Sonnet (Anthropic, 2024) and Gemini-1.5-Pro (Team et al., 2024)) and five open-source models (*i.e.*, DeepSeek-V3 (Liu et al., 2024a), DeepSeek-R1 (Guo et al., 2025), and Qwen2.5-14/32/72B-Instruct (Qwen Team, 2024)).

As shown in Table 3, the results indicate that state-of-the-art LLMs face significant challenges in coding tasks involving API updates. For example, leading commercial models like GPT-4o and Claude-3.5-Sonnet exhibit poor performance, with BLEU scores below 20% on the code completion task. Similarly, recently released models with up-to-date knowledge cutoffs, such as DeepSeek-V3 and DeepSeek-R1, which are expected to incorporate fresher code knowledge, also fail to accurately reflect API updates, yielding similarly low BLEU scores. These findings reveal systemic shortcomings in LLMs' ability to adapt to evolving APIs, highlighting the fundamental limitations of static pretraining paradigms. Thus, even the latest models suffer from knowledge decay as API versions evolve over time.

### 3.2. RQ2: Benchmarking Knowledge Updating Methods

We benchmark five knowledge updating methods including SFT-LoRA (Peng et al., 2023), DPO (Rafailov et al., 2023), SimPO (Meng et al., 2024), and ORPO (Hong et al., 2024), across five open-source LLMs including three code-specific LLMs (*i.e.*, CodeLlama-7B-Instruct (Roziere et al.,

Table 4: **The overall performance of different knowledge updating methods across five open-source LLMs.** We train five models using different methods and evaluate their performance on CODESYNCBENCH and HumanEval. All methods demonstrate limited effectiveness on CODESYNCBENCH. (BU for BLEU, RL for ROUGE-L, RED for Relative Edit Distance, and CBU for CodeBLEU.)

| Method | CCT | | | | ECT | | | | MCQ | | | HumanEval | |
|---|---|---|---|---|---|---|---|---|---|---|---|---|---|
| | BU↑ | RL↑ | RED↓ | CBU↑ | BU↑ | RL↑ | RED↓ | CBU↑ | P@1↑ | P@3↑ | P@5↑ | P@1↑ | Ratio↑ |
| *Qwen2.5-7B-Instruct* | | | | | | | | | | | | | |
| Original | 7.95 | 25.70 | 73.61 | 30.21 | 32.24 | 56.79 | 50.77 | 40.71 | 28.48 | 41.61 | 46.91 | **65.24** | – |
| SFT-LoRA | 12.17 | 34.59 | 68.76 | 32.32 | 26.63 | 44.81 | 57.15 | 42.85 | 32.83 | 47.55 | 53.21 | 62.80 | 96.26 |
| DPO | 24.45 | **52.94** | 57.12 | 39.24 | **46.24** | 64.87 | 42.99 | 49.75 | **33.39** | 45.61 | 50.05 | 61.59 | 94.41 |
| ORPO | **24.90** | 52.33 | **56.37** | 38.77 | 40.98 | 58.92 | 47.63 | 46.38 | 32.85 | **47.74** | **53.35** | 63.41 | 97.19 |
| SimPO | 24.81 | 52.90 | 56.88 | **39.67** | 45.15 | **65.51** | **42.90** | **51.02** | 33.14 | 44.35 | 48.69 | 63.41 | 97.19 |
| *Qwen2.5-Coder-7B-Instruct* | | | | | | | | | | | | | |
| Original | 5.89 | 21.56 | 76.58 | 29.41 | 11.64 | 26.78 | 71.81 | 32.68 | 32.56 | 41.28 | 44.57 | 82.32 | – |
| SFT-LoRA | 15.44 | 37.40 | 66.55 | 31.17 | 19.20 | 40.68 | 60.93 | 36.03 | 35.16 | 48.63 | **55.02** | 82.32 | 100.00 |
| DPO | 23.36 | 51.82 | 46.12 | 38.67 | 55.57 | 59.07 | 46.12 | 44.95 | 37.00 | 46.39 | 50.40 | **82.93** | **100.85** |
| ORPO | 21.47 | 48.17 | 53.43 | 37.06 | **56.92** | 50.20 | 53.43 | 40.62 | 35.42 | **48.64** | 54.70 | 81.71 | 99.26 |
| SimPO | **23.86** | **53.17** | **45.22** | **39.39** | 54.57 | **60.31** | **45.22** | **45.53** | **37.87** | 44.92 | 47.80 | **82.93** | **100.85** |
| *Llama-3.1-8B-Instruct* | | | | | | | | | | | | | |
| Original | 5.99 | 22.45 | 75.70 | 28.94 | 17.68 | 40.98 | 63.41 | 35.18 | 29.08 | **54.39** | **66.28** | 62.20 | – |
| SFT-LoRA | 13.21 | 36.70 | 72.01 | 34.54 | **43.78** | **65.76** | **41.84** | **49.90** | 22.28 | 38.74 | 47.24 | 60.98 | 98.04 |
| DPO | 24.13 | 51.36 | **55.38** | **38.85** | 27.18 | 51.57 | 54.83 | 38.88 | 36.42 | 49.88 | 55.34 | 58.54 | 94.12 |
| ORPO | 21.55 | 44.19 | 60.62 | 37.12 | 24.27 | 42.21 | 62.09 | 36.81 | 31.47 | 50.30 | 58.74 | 60.37 | 97.06 |
| SimPO | **26.83** | **53.95** | 56.07 | 36.79 | 23.04 | 44.91 | 58.74 | 39.69 | **36.56** | 43.96 | 46.66 | **62.20** | **100.00** |
| *CodeLlama-7B-Instruct* | | | | | | | | | | | | | |
| Original | 8.44 | 28.25 | 73.20 | 30.22 | 18.11 | 37.71 | 64.45 | 35.86 | 10.89 | 24.79 | 33.24 | **38.41** | – |
| SFT-LoRA | 17.24 | 44.97 | 59.57 | 34.36 | 30.60 | 50.42 | 53.99 | 43.51 | 10.34 | 18.91 | 24.85 | 36.59 | 95.26 |
| DPO | 26.54 | 53.27 | **26.51** | 41.66 | 39.67 | 60.55 | 44.79 | 48.15 | 20.48 | 41.09 | 51.71 | 36.59 | 95.26 |
| ORPO | 24.37 | 50.70 | 54.61 | 40.01 | 36.06 | 55.69 | 49.00 | 46.35 | 18.07 | 39.17 | 51.26 | 35.37 | 92.09 |
| SimPO | **27.78** | **56.48** | 50.62 | **42.04** | 40.56 | **65.27** | **41.65** | **49.08** | **25.40** | 45.50 | 54.66 | 35.98 | 93.67 |
| *DeepSeek-Coder-6.7B-Instruct* | | | | | | | | | | | | | |
| Original | 5.97 | 22.55 | 75.51 | 29.03 | 30.07 | 53.11 | 52.20 | 41.69 | **31.25** | 24.29 | 43.60 | **72.56** | – |
| SFT-LoRA | 14.96 | 41.42 | 62.45 | 33.34 | **47.79** | **71.25** | **34.32** | **53.76** | 7.88 | 8.89 | 9.32 | 71.34 | 98.32 |
| DPO | 26.77 | 55.72 | 50.86 | **42.54** | 43.29 | 64.95 | 41.91 | 50.35 | 6.37 | 8.61 | 9.00 | 70.12 | 96.64 |
| ORPO | **28.39** | **56.99** | **49.23** | 42.47 | 43.77 | 64.86 | 41.32 | 48.70 | 7.02 | 7.79 | 8.04 | 68.29 | 94.12 |
| SimPO | 25.10 | 53.69 | 52.97 | 41.50 | 41.47 | 64.06 | 42.50 | 50.08 | 6.75 | 9.21 | 10.55 | 68.29 | 94.12 |

2023), Qwen2.5-Coder-7B-Instruct (Hui et al., 2024), and DeepSeek-Coder-6.7B-Instruct (Guo et al., 2024a)) and two general-purpose LLMs (*i.e.*, Llama-3.1-8B-Instruct (Dubey et al., 2024) and Qwen2.5-7B-Instruct (Qwen Team, 2024)). Detailed experiment settings are listed in Appendix D.2.

**Evaluation of Updating Effectiveness.** As illustrated in Figure 2 and Table 4, the results indicate that knowledge updating methods can improve LLMs' performance in handling API evolution across the three evaluation tasks. Notably, fine-tuned LLMs with size 6.7B-8B can achieve scores comparable to those of leading proprietary and open-source LLMs, such as Claude-3.5-Sonnet, with BLEU scores of 23.86%-31.59 on the CCT task. Despite these improvements, the absolute scores remain low, indicating that current methods are insufficient for effectively updating the code knowledge of LLMs.

Notably, the DeepSeek-Coder-6.7B-Instruct model exhibits an anomaly on the MCQ task, where fine-tuning leads

to significantly lower scores. Analysis of the model outputs reveals degraded instruction-following capabilities, resulting in non-compliant responses. In contrast, other models maintain compliant outputs, indicating a lack of robustness in this model.

Overall, while fine-tuning narrows the gap with larger models in some cases, the persistently low scores reveal the limitations of existing approaches. Further advances (i.e. integrating structural code understanding or continual learning) are required to more reliably update LLMs' code knowledge without compromising their general capabilities.

**Fine-grained Analysis on Qwen.** Table 4 demonstrates that the evaluated models suffer from severe knowledge obsolescence issues. To assess models' intrinsic capabilities on code, we construct a variant of the CCT benchmark where the reference answer corresponds to **outdated** code knowledge. Performance on this variant serves as an upper-bound estimate of models' code knowledge. As

Table 5: **Estimation of the Upper-Bound Performance on code knowledge.**

| Model | Original | Best Method | Upper Bound |
|---|---|---|---|
| Qwen2.5 | 30.21 | 39.67 | **42.05** |
| Qwen2.5-Coder | 29.41 | 39.39 | **45.12** |

Table 6: **Performance of RAG Baseline.**

| Method | CCT CBU↑ | ECT CBU↑ | MCQ CBU↑ |
|---|---|---|---|
| Original | 29.41 | 32.68 | 32.56 |
| SFT | 31.17 | 36.00 | 35.16 |
| DPO | 38.67 | 44.95 | 37.00 |
| RAG | 35.17 | 42.26 | 34.26 |
| SFT+RAG | **40.70** | **51.35** | **36.89** |

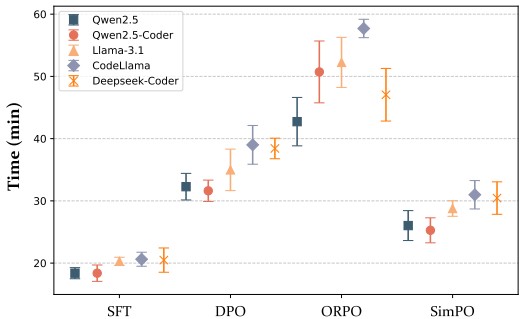

Figure 5: **Efficiency of different knowledge updating techniques.** We measure and compare the time consumption of four knowledge updating techniques across five models. We can observe that the training durations follow the pattern: SimPO < DPO < ORPO.

shown in Table 5, we select Qwen2.5-7B-Instruct and Qwen2.5-Coder-7B-Instruct for evaluation on CCT variant using CodeBLEU metric. The results indicate that current techniques fall short of the upper bound, underscoring the limitations in their effectiveness.

Furthermore, we introduce retrieval-augmented generation (RAG) (Lewis et al., 2020) as the additional baseline. We construct a vector database to store all API signatures from the target library with text-embedding-3-large (OpenAI) as the embedding model. Table 6 reports the performance of Qwen2.5-Coder-7B-Instruct on CODESYNCBENCH using CodeBLEU as the metric. Across three different tasks, RAG performs better than SFT but still falls short of DPO. The relatively limited performance can be attributed to its hit rate of only 60%. This reduced hit rate is largely caused by the presence of many similarly named APIs and the complexity introduced by the large number of APIs present in the code context. Notably, combining SFT and RAG achieves improved performance, demonstrating the potential benefits of integrating external retrieval with fine-tuning.

**Evaluation of Updating Efficiency.** In addition to effectiveness, updating efficiency is a crucial factor that may influence developers' adoption in practice. For each model, we recorded the training time required for four knowledge updating methods, as shown in Figure 5. The results indicate that SFT-LoRA is the most efficient method overall. Moreover, we can observe that, across all models, the training durations follow the pattern: SimPO < DPO < ORPO, indicating that ORPO is the least efficient and SimPO is the most efficient. Additionally, it can be seen that the training duration for ORPO exhibits relatively larger fluctuation, indicating instability in efficiency.

**Evaluation of Model Utility Post-Updating.** We evaluate the general utility of the LLMs before and after updating using the widely used HumanEval benchmark (Chen et al., 2021b). For each problem, we sample 10 answers (*i.e.*, $n = 10$) and calculate Pass@1, Pass@3, and Pass@5 scores. To assess the impact of updating, we computed the **ratio** of the Pass@5 scores for models trained with various methods to those of the original model. The results show that most updating methods incurred a score loss of no more than 10%, indicating a minor impact on the models' overall utility.

### 3.3. RQ3: Impact of API Updating Settings

We further investigate the impact of different API update settings such as the numbers of API invocations available for training and the types of updated APIs, on the performance of knowledge updating in API evolution tasks.

**Impact of Update-Aware Instruction Number.** To evaluate this, we filter 32 APIs from the original training set, each with more than 50 invocation samples, and construct four new training sets with 5, 10, 20, and 50 samples per API, respectively. We then train Qwen-2.5-7B-Instruct using four knowledge updating techniques (*i.e.*, SFT-LoRA, DPO, ORPO, SimPO) on these sets and evaluate performance on the code completion task. As shown in Figure 6, using only 5 samples per API results in relatively poor performance. When the training sample number increases to 10 per API, the model demonstrates improved recall capabilities of the updated APIs. Further increases in sample number lead to performance stabilization with minor additional gains. These findings suggest that a moderate number of samples is sufficient for LLMs to internalize new code knowledge, with 10 samples per API striking an optimal balance between effectiveness and efficiency.

**Impact of Updated API Type.** We evaluate Qwen-2.5-7B-Instruct on the CCT task across different API types. As illustrated in Figure 7, a clear trend can be observed among the three API types. The knowledge updating methods perform similarly on function APIs and initializer APIs yet exhibit significantly lower performance on method APIs. This discrepancy can be attributed to the intrinsic complexity

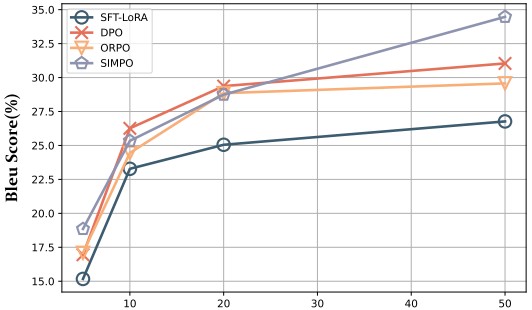

Figure 6: **Model performance with varying numbers of invocation instances per API.** We divide the original training set into subsets containing different numbers of samples per API (5, 10, 20, 50). The Qwen2.5-7B-Instruct is trained on these subsets and evaluated on the Code Completion Task. The result indicates that 10 samples per API is sufficient for injecting knowledge, keeping a balance between performance and efficiency.

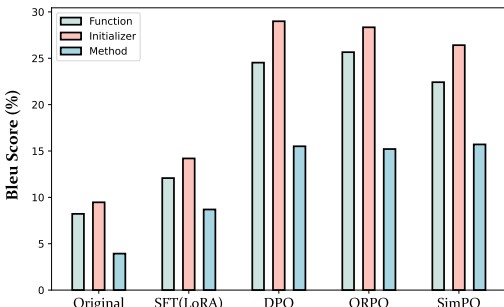

Figure 7: **Model performance on different types of APIs.** We evaluate the performance of Qwen-2.5-7B-Instruct, trained using various techniques, as well as a reference model, on different categories of APIs (functions, methods, and initializers). The results reveal significant differences in the models' capabilities across different categories. Notably, all models perform relatively worse on methods compared to functions and initializers.

of method invocations, which typically involve class instantiations, object references, and dynamic method calls. Unlike function and initializer APIs that follow relatively straightforward invocation patterns, method APIs require LLMs to correctly infer object types, track dependencies, and manage class hierarchies. These additional layers of complexity increase the difficulty of accurately invoking API updates, making it more challenging for LLMs to learn and apply correctly. Addressing these challenges may require more sophisticated knowledge updating strategies to improve LLMs' adaptability to complex code knowledge.

## 4. Related Work

**LLMs for Code Generation.** Both proprietary (OpenAI, 2024; Team et al., 2024) and open-source LLMs (Hui et al., 2024; Roziere et al., 2023; Guo et al., 2024a) have recently demonstrated strong code generation abilities, leading to AI-driven tools such as Copilot (GitHub, 2024) and Cursor (AI, 2024). However, these models often overlook the risks associated with outdated APIs. Existing benchmarks and studies either rely on synthetic API updates (Liu et al., 2024c) or vaguely defined knowledge-editing tasks (Li et al., 2024b), limiting their applicability. Our work addresses these gaps by benchmarking knowledge updating methods on real-world API changes.

**Knowledge Updating for LLMs.** LLMs are prone to knowledge obsolescence, as retraining is computationally expensive. Knowledge updating methods (i.e. supervised fine-tuning (Liu et al., 2024c; Peng et al., 2023), reinforcement learning (Schulman et al., 2017; Meng et al., 2024; Rafailov et al., 2023; Hong et al., 2024), and

knowledge model editing (KME) (Meng et al., 2022a; Hartvigsen et al., 2023; Meng et al., 2022b)) aim to efficiently integrate new information. KME methods optimize specific neurons related to new knowledge with no performance degradation.

**Data Synthesized by LLMs.** LLMs are widely used to generate synthetic data for pretraining and fine-tuning (Liu et al., 2024b), covering diverse applications like multilingual QA (Riabi et al., 2021), chatbot conversations (Zhao et al., 2023; Zhang et al., 2024b), and data augmentation (Dai et al., 2025; Chung et al., 2023; Chen et al., 2024a; Pu et al., 2025; Huang et al., 2025). Synthetic benchmarks further require generated data to be diverse, accurate, and challenging (Chen et al., 2025; Wu et al., 2024), and are now used to evaluate emergent capabilities, such as trustworthiness (Huang et al., 2024; Ye et al., 2024; Gao et al., 2024) and multimodal reasoning (Zhang et al., 2024a; Bao et al., 2024; Chen et al., 2024b; Fu et al., 2025). We advance this area by proposing a synthetic benchmark integrating three challenging code generation tasks.

## 5. Conclusion

In this paper, we introduce CODESYNC, an innovative data engine for constructing the structured benchmark CODESYNCBENCH, to evaluate LLMs' ability in handling evolving code knowledge. Benchmarking the state-of-the-art LLMs and popular knowledge update techniques, we find that LLMs struggle with rapid API evolutions. Furthermore, existing techniques are insufficient for effective code knowledge integration. This highlights the necessity for improved approaches to help models adapt to evolving code knowledge in dynamic environments.

## Acknowledgements

This work is partially supported by the Major Program (JD) of Hubei Province (Grant No. 2023BAA024). Dongping Chen and Yao Wan are supported by the Fundamental Research Funds for the Central Universities (HUST: 62400001). We would like to thank all the anonymous reviewers for their insightful comments.

## Impact Statement

In this paper, we present CODESYNC, an innovative data engine designed to systematically monitor real-world API changes and generate CODESYNCBENCH, a specialized benchmark for assessing and improving LLMs' adaptability to API updates. This benchmark establishes a standardized evaluation framework for assessing the challenges posed by outdated API knowledge in LLMs. However, one limitation of our work is the efficiency of collecting invocation instances. By enabling LLMs with real-time API adaptation capabilities, our work has the potential to significantly enhance developer productivity and drive advancements in software development, AI-driven coding assistants, and programming education.

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

## A. Comprehensive Related Works

**Deep Learning for Code Intelligence.** Neural language models have made remarkable progress in code intelligence (Wan et al., 2024), encompassing a variety of tasks including code summarization (Wan et al., 2018; Wang et al., 2020), code search (Gu et al., 2018; Wan et al., 2019), and code generation (Bi et al., 2024; Sun et al., 2024; Li et al., 2024a). A central challenge in code intelligence is the effective representation of source code as vectors. Substantial effort has been devoted to this, primarily through the design of deep neural networks in three main categories: sequential code tokens (e.g., plain text, intermediate representations, APIs), Abstract Syntax Trees (ASTs), and code graphs (such as control-flow graphs, data-flow graphs, and code property graphs). For sequential code tokens, approaches have employed Recurrent Neural Networks (RNNs) (Graves & Graves, 2012; Chung et al., 2015; Gu et al., 2018) and Convolutional Neural Networks (CNNs) (Mou et al., 2016; Yamashita et al., 2018) to process plain text (Iyer et al., 2016; Allamanis et al., 2016), intermediate representations (VenkataKeerthy et al., 2020; Peng et al., 2021; Gui et al., 2022), and API calls (Gu et al., 2016; Nguyen et al., 2017) extracted from source code. For ASTs, prior research has either developed structural RNNs (Wan et al., 2018) and CNNs (Mou et al., 2016) to capture the hierarchical structure of the tree or linearized the AST into sequential traversals (Alon et al., 2019; 2018) for processing with traditional RNNs or CNNs. To handle code graphs, various Graph Neural Networks (GNNs) (Chu et al., 2024; Allamanis et al., 2018) have been proposed, enabling more sophisticated representations of code structure and semantics. Recently, advancements in LLMs for text generation have spurred the emergence of specialized code-focused LLMs, including CodeT5+ (Wang et al., 2023), InCoder (Fried et al., 2023), StarCoder (Li et al., 2023), Code Llama (Roziere et al., 2023), WizardCoder (Luo et al., 2024), Qwen-Coder (Hui et al., 2024), and DeepSeek-Coder (Guo et al., 2024a). Despite recent advances, LLMs still struggle to keep pace with rapidly evolving programming knowledge. This paper explores methods for integrating dynamic knowledge, enabling LLMs to synchronize with the ongoing developments in programming languages, frameworks, and best practices.

**LLMs for Code Generation.** Recently, LLMs such as the commercial/black-box GPT-4 (OpenAI, 2024), Gemini (Team et al., 2024), and open-source models like Qwen-Coder (Hui et al., 2024), Code Llama (Roziere et al., 2023), and DeepSeek-Coder (Guo et al., 2024a), have demonstrated impressive capabilities in generating high-quality code. Building on these LLMs, several products, including Copilot (GitHub, 2024) and Cursor (AI, 2024), have been developed. However, the security risks posed by outdated APIs are often overlooked, and existing studies on the code knowledge update task have significant limitations. For example, the benchmark proposed by Liu et al. (2024c) generates API update pairs by prompting ChatGPT (OpenAI, 2024) rather than collecting authentic APIs. Li et al. (2024b) construct an instruction benchmark where the subject and object of knowledge are vaguely defined, but apply knowledge model editing techniques to model tuning. In this paper, we aim to benchmark knowledge updating methods for real-world API updates using authentic GitHub releases.

**Knowledge Updating for LLMs.** LLMs often rely on data from a specific time period, leading to outdated knowledge that retraining can not easily fix due to its high computational cost. To address this, knowledge updating techniques offer a more efficient way to integrate new information without sacrificing the model's current capabilities. One approach is supervised fine-tuning (SFT) (Liu et al., 2024c; Peng et al., 2023), which optimizes model parameters to integrate new knowledge directly. Other methods treat new knowledge as preferred behavior over outdated information, such as reinforcement learning from human feedback (RLHF) methods (Schulman et al., 2017; Meng et al., 2024; Rafailov et al., 2023; Hong et al., 2024), which is efficient for refining model behavior to align with new knowledge. Knowledge neuron theory (Dai et al., 2021) takes a further step by formulating knowledge as a tuple $\{s, r, o\}$, where $s$, $r$, and $o$ represent the **subject**, **relation**, and **object** of knowledge, respectively. Based on this, knowledge model editing (Meng et al., 2022a; Hartvigsen et al., 2023; Meng et al., 2022b) emerge as a more cost-effective and time-efficient approach for updating knowledge. These methods first identify key neurons linked to the new knowledge and then optimize them, carefully preserving the language model's overall capabilities. However, Li et al. (2024b) reveal that many KME techniques struggle with effectiveness and fail to generalize.

**Data Synthesized by LLMs.** LLMs have demonstrated an impressive capacity for data generation, leading to their application in creating synthetic datasets for pretraining and finetuning, replacing the labor-intensive processes of manual data scraping and selection (Liu et al., 2024b). Distinct from earlier methods that focus on traditional language models (Schick & Schütze, 2021), LLMs offer enhanced prospects for producing high-quality synthetic data across a wide spectrum of applications, such as multilingual QA (Riabi et al., 2021), chatbot conversation (Zhao et al., 2023; Zhang et al., 2024b), and data diversity augmentation (Dai et al., 2025; Chung et al., 2023; Chen et al., 2024a). The concept of synthetic benchmarks takes a step further by demanding that the LLM-generated data be diverse, accurate, and systematically challenging (Chen et al., 2025; Wu et al., 2024). Moreover, synthetic benchmarks have also been constructed in evaluating

LLM emergent capabilities such as trustworthiness (Huang et al., 2024; Ye et al., 2024; Gao et al., 2024), persona-based conversation (Jandaghi et al., 2023), and multimodal domain (Zhang et al., 2024a; Bao et al., 2024; Chen et al., 2024b). Our research advances a synthetic benchmark for code generation by developing a paradigm that integrates three challenging code generation tasks. Recently, in response to concerns about the quality of synthetic datasets, Dekoninck et al. (2024b) conduct comprehensive experiments to evaluate the diversity and fidelity of synthetic data produced by LLMs, while Dekoninck et al. (2024a) introduce a new inference framework, model arithmetic, to control the generated content.

## B. Detailed Experiment Setups

### B.1. Dataset

#### B.1.1. API COLLECTION

The initial step of CODESYNC pipeline involves collecting APIs from various libraries. To achieve this, we utilize the Python built-in module *inspect*, which enables us to navigate through library files and compile a comprehensive list of all available APIs. In this part, we will delve into the detailed process of how to collect APIs comprehensively from libraries.

**C-extension APIs.** C-extension methods and functions are a powerful feature in Python programming that are employed in many third-party libraries, (*e.g.*, NumPy, PyTorch), to accelerate execution efficiency. One of the key feature of C-extension functions and methods is their support for function overloading. Function overloading allows a single API name to be used with multiple different parameter lists, or signatures. This means to collect various versions of signatures for each API.

**Inspect Module.** Python built-in module, *Inspect*, provides several useful functions for introspecting live objects, such as functions, classes, and modules. It allows us to retrieve information about source code of Python objects, such as signature, arguments and documentation.

**Categories.** Python offers a diverse range of APIs, each designed for specific purposes and governed by distinct invocation rules. In this study, we focus on three primary types: function APIs, method APIs, and initializer APIs. These categories not only highlight Python's core capabilities but also exhibit unique characteristics and behaviors. Function APIs are standalone entities that can be invoked without requiring a class or instance context. In contrast, method APIs are inherently tied to class instances, leveraging encapsulation and object-oriented programming principles. The invocation rules for methods differ significantly from those for functions, reflecting their object-oriented nature. Additionally, Python provides several magic methods that are denoted by double underscore ('__') at the beginning and end of their names. Among these, initializers (*i.e.*, '__init__') are the most commonly used, serving as a method for object creation and initialization. To evaluate and benchmark Python APIs evolution comprehensively, we select representatives from these three categories to construct our benchmark CODESYNCBENCH.

#### B.1.2. IDENTIFYING API UPDATES

**Multiple Types of Parameter.** The three fundamental types of parameters are *positional-only parameter*, *keyword-only parameter* and *positional & keyword parameter*. The term 'positional' refers to parameters that can be passed only according to its position in definition. 'Keyword' is the name of parameter in the function signature, allowing passing parameter with marking it explicitly instead of position. There are two special symbols in API signatures (*e.g.*, *, /). Parameters set before '*' are **positional-only parameters**, which must be passed in order according to theirs positions in definition, and parameters located after '/' are **keyword-only parameters**, requiring marking parameter name when used; otherwise, a syntax error will occur. Additionally, parameters can be also categorized according to default values into 2 types, **required parameters** and **optional** parameters. Therefore, changes of parameter types have impact on invocation rules, which should be considered when determining API update operations.

**API Update Determination.** How to determine API update operations? The most straightforward changes include the addition or deletion of parameters. A more nuanced level of analysis involves examining changes in parameter types as these alterations can significantly impact the rules for invoking APIs. Therefore, API updates can be categorized into 2 primary aspects, **the addition of deletion of parameters** and **changes in parameter types**. To effectively identify API updates, it is crucial to focus on parameter changes, including both the mapping relationships between parameters and modifications to their types. To systematically capture these changes, we construct **parameter mappings** for each pair of APIs, establishing connections between corresponding parameters in the outdated and latest version of their signatures. Specifically, parameter

mapping enables categorize two distinct aspects. First, if a parameter mapping can be successfully constructed , it implies that all parameters are consistently present in both versions of signatures, indicating no additions or deletions. Following this, the next step involves a detailed examination of each parameter pair within mappings, focusing on comparing their attributes to identify any modifications or differences. This approach enables a clear and structured understanding of how APIs involve over time.

**Parameter Renaming.** Static analysis, however, has inherent limitations, especially in cases where parameter renaming occurs. It is challenging to infer changes in functionality solely based on parameter names. For example, in `transformers==4.47.0`, the API `transformers.pipelines.get_task` has a parameter named `use_auth_token`, whereas the keyword of this parameter was `token` in version `transformers==4.25.1`. In spite of the same functionality, renaming makes it impossible to recognize their equivalence solely by analyzing signatures. In this process, we assume that keywords of parameters are strongly connected to their functionality. The similarity between keywords suggests the similarity of their functionality. Instead of excluding all of name modification situations, we first set a threshold and compute the keyword similarity scores to account for some simple modifications. Based on this, we will then construct parameter mapping according to keyword mappings for further explorations.

**Establishing Parameter Mappings.** However, the inherent complexity of Python API signatures poses significant challenges in accurately establishing parameter mappings. To address this, we establish three rules that must be satisfied to determine whether no modification has occurred. Python introduces two special symbols ('/' and '*'), which divide parameters into three categories, **positional-only**, **keyword-only** and **positional-and-keyword** parameters. Specifically, we construct three individual parameters mappings for these types of parameters and establish three rules that must be satisfied to determine whether no modification has occurred.

- **Rule 1: Successful Parameter Mapping.** A valid parameter mapping must be constructed, ensuring that both the number of parameters and their corresponding keywords remain identical across different signatures.
- **Rule 2: Type-Specific Consistency.** Each parameter type must follow specific rules:
  - For **positional-only** parameters, the order of parameters in the function definition must remain strictly unchanged across signatures.
  - For **keyword-only** parameters, the parameter names (keywords) must remain consistent to preserve their correspondence.
  - For **positional-and-keyword** parameters, both the order requirement and keyword consistency must be satisfied simultaneously.
- **Rule 3: Required vs. Optional Parameters.** Parameters can be further categorized into two types: **required** parameters, which must be provided when invoking APIs, and **optional** parameters, which have default values. While revisions to default values are not considered API updates, the type of a parameter must remain unchanged.

These rules collectively provide a practical methodology for evaluating parameter modifications and determining API consistency, which is a crucial part of CODESYNC implementing completely autoamted pipeline.

### B.1.3. API INVOKING INSTANCES CRAWLING

After obtaining updated APIs along with corresponding information, it is necessary to crawl API invocations from ground truth which will be used to inject API knowledge into LLM for further exploration. Actually, directly feeding signature to models for tuning is unlikely to be effective, and limited to reflect comprehensive information, such as invoking rules, which is hard to be formulated. Therefore, we collect a large dataset of invocation instances to implicitly reflecting relative knowledge.

**Real-World API Invocation.** Synthesizing invocation completely relied on LLM is a convenient method for constructing dataset. However, this method exists inherent limitations. For example, information implied in context of generated code is insufficient and the contextual scenario is restricted to LLMs' embedded knowledge. The inevitable bias therefore poses challenges to comprehensively reflect authentic invoking rules and habits. Instead of synthesizing invocations, we try to crawl code from GitHub with the help of GitHub Code Search, a Code Search Engine developed by GitHub to effectively aggregate repositories or files using regular expression. Additionally, We involve **search templates** as shown in B.1.3, to enhance the effectiveness of invocation retrieval

**Search Templates.** Python allows aliases declaration of import statements to simplify usage of third-party modules and APIs. In the authentic programming scenario, directly invoking APIs with full name fails to align with developers'

programming habits. We therefore design a set of templates for each library to expand searching scope. For example, while the module `torch.nn.functional` is imported, these statements might exist:

> 1. import torch.nn.functional as F
>
> 2. from torch.nn import functional as F

For any field in the API name (a segment separated by dots), an alias can be assigned and there are two formats: `import as` and `from import`. Based on these characteristics, we can generate a series of searching templates. Templates of `torch.nn.functional.softmax` are shown as below:

> 1. "torch.nn.functional.softmax" (directly match)
>
> 2. "import torch as" + ".nn.functional.softmax"
>
> 3. "from torch import nn" + ".functional.softmax"
>
> 4. "import torch.nn as" + ".functional.softmax"
>
> 5. "from torch.nn import functional" + ".softmax"
>
> 6. "import torch.nn.functional as" + ".softmax"
>
> 7. "from torch.nn.functional import softmax"

In the second template, we match "import torch as" instead of "import torch". This is because when the module is imported without an alias (e.g., simply "import torch"), the full path "torch.nn.functional.softmax" will be directly used in the code. For **function** APIs and **initializer** APIs, the above patterns can be directly applied for decomposition. We next utilize GitHub Code Search to retrieve code that contains all segments for each template (with an upper limit of 500 files).

Different from function and initializer, **method** APIs requires a further step due to dynamic binding mechanism. A method API can be divided into two parts: **class name** and **method name**. For example, `torch.Tensor` and `shape` are class name and method name of `torch.Tensor.shape`, respectively. In the most programming scenario, Python objects lack explicit type definitions. To align with subsequent procedures, we only take one specific situation into consideration where both type declaration and API invocation exist in the same file simultaneously. Searching templates can be applied on method APIs retrieval as well, while an additional segments, `f".{method name}("`, should be included. For API `torch.Tensor.shape`, each template will include `".shape("`. Explicit type declarations will be clarified in Appendix B.1.4.

### B.1.4. LOCATING VALID API INVOCATIONS

After retrieving a dataset of files that contain relative substring of target API invocation, further filtering is required to identify code that genuinely invokes the target API. The following illustration is divided into two parts: **function / initializer APIs locating** and **method APIs locating**.

**Function / Initializer APIs Locating.** Initializer APIs share similar invoking rules with those of function APIs. We can use abstract syntax tree(AST) to analyze crawled files for locating the target API invocations. Specifically, this part contains two steps: **(1) Alias Mapping**: We scan the import statements and construct mappings between original library/module name and aliases. **(2) Invocation Analysis**: Based on alias mapping, we traverse the AST of files and analyze each invocation statement to determine whether the target API are invoked. The start & end line number of invocations will be recorded for subsequent process.

**Method APIs Locating.** Invocations of method APIs are often associated with class instances. To determine method API invocations, we need to infer the types of variables that invoke the methods. However, variables are dynamically bound to types during program execution. We therefore focus on situations where the types of variables can be statically inferred from the raw code. There are three situations:

- Variables are assigned by using initializer of target class.
- Type annotations are provided in function definitions.
- Function definitions provide return type annotations.

The first step is to scan the whole file to record types of variables as well as their scopes. We next traverse the AST, tracking target class instances in their own scope to identify methods they invoked.

**Format Conversion.** After locating and recording API invocations in each file, we perform two operations to split the data: **(1) Segment Split**: Treating the entire file as a single dataset item is inefficient and redundant.To better utilize the crawled files, we split each file into multiple segments based on function definition. In other words, each segment corresponds to a complete function definition and is treated as an individual dataset item. **(2) Metadata Convert**: Each segment is then further divided into three parts: **code context**, **target sequence** and **code suffix**. The code context is the prompt in subsequent tasks. To avoid knowledge leaking, the target sequence is the first invocation of target API within the segment. These split operations allow for more efficient processing and better representation of the code's structure, ultimately improving the dataset's usability for subsequent tasks.

## B.2. Models

**Qwen-2.5-7B-Instruct.** A 7-billion parameter instruction-tuned model designed for general-purpose tasks, offering robust performance across various applications by following user instructions effectively.

**Qwen-2.5-Coder-7B-Instruct.** A specialized 7-billion parameter model tailored for coding-related tasks, excelling in code generation, debugging, and understanding programming languages through instruction-following capabilities.

**Llama-3-8B-Instruct.** An 8-billion parameter instruction-tuned model built for versatile applications, providing strong performance in natural language understanding and task execution based on user instructions.

**CodeLlama-7B-Instruct.** A 7-billion parameter model fine-tuned for coding tasks, optimized for generating, analyzing, and refining code while adhering to user-provided instructions.

**DeepSeek-Coder-6.7B-Instruct.** A 6.7-billion parameter model specifically designed for coding and programming tasks, leveraging instruction-tuning to deliver accurate and efficient code-related solutions.

## B.3. Knowledge Updating Methods

### B.3.1. DIRECT PREFERENCE OPTIMIZATION (DPO)

Traditional reinforcement learning algorithms (*e.g.*, PPO (Schulman et al., 2017)) introduce reward models to guide LLMs to align with human preferences. While these methods exhibit superior performance in many fields, they suffer from extremely high computational costs and require a large amount of training data to optimize policy of reward models. To accelerate the process of training, DPO directly optimizes the model's policy to align with human preferences by leveraging pairwise comparison data. Each data pair consists of a preferred sample $\mathbf{y}_i^+$ and a dispreferred sample $\mathbf{y}_i^-$ for a given input $\mathbf{x}_i$. DPO adjusts the model to increase the likelihood of generating preferred outputs while reducing the probability of dispreferred ones. By implicitly encoding preference rankings into the objective function, DPO eliminates the need for explicit reward modeling or complex reinforcement learning pipelines, offering a simpler and more stable training framework.

The key insight of DPO is to reframe preference learning as a supervised likelihood optimization problem. Given preference pairs $(\mathbf{x}_i, \mathbf{y}_i^+, \mathbf{y}_i^-)$, the objective maximizes the log-likelihood difference between preferred and dispreferred outputs:

$$\mathcal{L}_{\text{DPO}} = \sum_i \log \sigma \left( \log \frac{\pi_\theta(\mathbf{y}_i^+|\mathbf{x}_i)}{\pi_{\text{ref}}(\mathbf{y}_i^+|\mathbf{x}_i)} - \log \frac{\pi_\theta(\mathbf{y}_i^-|\mathbf{x}_i)}{\pi_{\text{ref}}(\mathbf{y}_i^-|\mathbf{x}_i)} \right),$$

where $\sigma$ denotes the sigmoid function and $\pi_{\text{ref}}$ represents the reference policy. This formulation ensures the model assigns higher probabilities to preferred responses relative to the reference policy while maintaining generation diversity through implicit regularization.

### B.3.2. ODDS RATIO PREFERENCE OPTIMIZATION (ORPO)

ORPO introduce *Odd Ratio* to quantify the preference learning. Specifically, it enhances preference learning by explicitly optimizing the odds ratio between preferred and dispreferred responses. The loss function combines log-odds maximization

with KL-divergence regularization:

$$\mathcal{L}_{\text{ORPO}} = \sum_i \log \frac{\pi_\theta(\mathbf{y}_i^+|\mathbf{x}_i)}{\pi_\theta(\mathbf{y}_i^-|\mathbf{x}_i)} - \lambda \cdot \text{KL}\left(\pi_\theta \| \pi_{\text{ref}}\right),$$

where $\lambda$ controls the regularization strength. This dual objective encourages preference alignment while preventing excessive deviation from the reference policy, addressing the exploration-exploitation trade-off inherent in policy optimization. ORPO's probabilistic framing improves sample efficiency in low-data regimes and enhances robustness to noisy preference labels.

### B.3.3. SIMPLE POLICY OPTIMIZATION (SIMPO)

SimPO extends the paradigm of DPO through architectural simplifications that enhance both training efficiency and alignment precision. At its core, SimPO reinterprets the alignment task as a margin maximization problem, where the model learns to maintain a specified quality gap between preferred and dispreferred responses. This is achieved through two synergistic mechanisms:

**Dynamic Length Normalization**: Traditional probability-based rewards inherently favor longer sequences due to multiplicative probability chains. SimPO counteracts this bias by computing rewards as *length-normalized* token probabilities:

$$R_\theta(y|x) = \frac{\beta}{|y|} \sum_{t=1}^{|y|} \log \pi_\theta(y_t|x, y_{<t}),$$

where the normalization factor $|y|$ (response length) ensures equal contribution per token, preventing length-based reward inflation. This design choice proves critical in tasks requiring concise yet high-quality responses, such as technical question answering or summarization.

**Adaptive Margin Enforcement**: Rather than relying on fixed hyperparameters, SimPO implements an intelligent margin threshold $m$ that interacts with the reward difference $\Delta R_\theta = R_\theta(y^+|x) - R_\theta(y^-|x)$:

$$\mathcal{L}_{\text{SimPO}} = \sum_i \max\left(0, m - \Delta R_\theta(x_i)\right).$$

The margin mechanism creates three distinct learning phases:

1. *Active Learning*: When $\Delta R_\theta < m$, gradients actively push the model to widen the reward gap

2. *Saturation Control*: Once $\Delta R_\theta \geq m$, gradient flow ceases to prevent over-optimization

3. *Implicit Regularization*: The margin $m$ automatically scales with batch statistics, adapting to varying preference strengths

By eliminating reference policy computations and reward modeling, SimPO achieves faster convergence while maintaining competitive performance. The margin-based objective automatically suppresses gradient updates when preference distinctions become clear, preventing overoptimization and reducing computational overhead. This makes SimPO particularly effective for aligning LLMs with limited computational resources.

# C. Prompts

## C.1. Prompt to Update Code Legacy

I will provide a code snippet as the context, followed by a calling statement that contains a target API call and a suffix. Additionally, the latest and outdated function signatures of the API are accessible(referred to as latest_signature and outdated_signature). Your task is to update the calling statement according to both the latest and outdated API function signatures, producing two distinct answers: the "latest answer" and the "outdated answer".

—

You must adhere to the following guidelines:
1. Calling Statement Updates: Only update the calling statement based on the given signatures, ensuring the functionality and correctness of the calls.
2. Include Required Parameters: The updated calling statements should include only the required parameters from the API signatures. Optional parameters should only be included if they are explicitly used or necessary based on the provided code context.
3. Avoid Unnecessary Defaults: Do not include default values for optional parameters unless they are explicitly mentioned in the code or are necessary for functionality.
4. Reflect API Updates: Clearly showcase the differences between the latest and outdated API signatures through your modifications.

—

Latest API Signature: [updated_signature]
Outdated API Signature: [outdated_signature]
Context: [context]
Statement: [target_seq]
suffix: [suffix]

## C.2. Prompt to Generate Wrong Choices for MCQ

I want to create a multiple-choice question where, based on a specific code context, we identify the most appropriate parameter list for the target API. I will provide you with the following information:

- `API_path`: The full name of the API
- `updated_signature`: The API's new signature
- `outdated_signature`: The API's old signature
- `import`: The import statements in the code
- `context`: The preceding code context, ending with the target API's name
- `updated_code`: The correct answer that matches the new signature
- `outdated_code`: The incorrect answer that matches the old signature

I want to construct a multiple-choice question with four options. Among these, `updated_code` will be the correct option, and `outdated_code` is one incorrect option I have already provided. You need to create two additional incorrect options based on the differences between the new and old signatures—specifically, options that would be "misleading" if a model is still relying on the old signature. In other words, if the model only knows the old signature, it might be inclined to select these incorrect answers.

Here are four possible approaches for crafting these additional incorrect options:

1. Remove some optional parameters from the correct answer (that is, `updated_code`).
2. Add some incorrect optional parameters, such as parameters that existed in the old signature but do not exist in the new one, or parameters that appear in neither signature (the name of these parameters should not be like `extra_param`, which can be judged to error very easily).
3. Rearrange the positions of any positional parameters based on `updated_code`.
4. Change parameter names, for example changing `add(x: int)` to something like `add(z=3)`.

**WARNING**: Your two new incorrect options MUST differ from **both** `updated_code` and `outdated_code` that I give to you, as well as from EACH OTHER.

**Output Format:**

Provide your two new incorrect options as your answer, **without** any other output.

For example:

```
############ Your output ##############

Option 1: (paramA, paramB=123)

Option 2: (paramX="hello")

######################################
```

—

API_path: [API_path]
updated_signature: [updated_signature]
outdated_signature: [outdated_signature]
import: [import]
context: [context]
updated_code: [updated_code]
outdated_code: [outdated_code]

# D. Experiment Settings

## D.1. Metrics

### D.1.1. BLEU METRIC

The BLEU score is used to evaluate the quality of generated text by comparing it to one or more reference texts. It is based on the precision of $n$-grams (contiguous sequences of words) in the generated text, with a brevity penalty to penalize overly short outputs. The BLEU score is calculated as follows:

$$\text{BLEU} = BP \cdot \exp\left(\sum_{n=1}^{N} w_n \log p_n\right),$$

where

- $BP$ is the **brevity penalty**, defined as:

$$BP = \begin{cases} 1 & \text{if } c > r \\ e^{(1-r/c)} & \text{if } c \le r. \end{cases}$$

  Here, $c$ is the length of the candidate (generated) text, and $r$ is the length of the reference text.
- $p_n$ is the **n-gram precision**, calculated as:

$$p_n = \frac{\text{Number of matching n-grams in candidate and reference}}{\text{Total number of n-grams in candidate}},$$

- $w_n$ is the weight for the $n$-th n-gram precision, typically set to $\frac{1}{N}$ for uniform weighting.
- $N$ is the maximum $n$-gram order (usually 4 for BLEU-4).

The BLEU score ranges from 0 to 1, where 1 indicates a perfect match with the reference text and 0 indicates no overlap with the reference text.

### D.1.2. ROUGE METRIC

The ROUGE metric is used to evaluate the quality of generated text by comparing it to one or more reference texts. It focuses on recall, measuring how much of the reference text is captured by the generated text. ROUGE has several variants, including ROUGE-N ($n$-gram overlap), ROUGE-L (longest common subsequence), and ROUGE-W (weighted longest common subsequence). In our experiments, we use ROUGE-L as the metric.

The ROUGE-L score is based on the longest common subsequence (LCS) between the candidate and reference texts. It is defined as:

$$\text{ROUGE-L} = \frac{\text{LCS}(C, R)}{\text{Length}(R)},$$

where

- $\text{LCS}(C, R)$ is the length of the longest common subsequence between the candidate text $C$ and the reference text $R$.
- $\text{Length}(R)$ is the length of the reference text.

The ROUGE score ranges from 0 to 1, where 1 indicates that the candidate text perfectly captures the reference text and 0 indicates no overlap with the reference text.

### D.1.3. RELATIVE EDIT DISTANCE METRIC

The Relative Edit Distance (RED) is a normalized metric used to measure the dissimilarity between two strings. It is calculated as the edit distance (e.g., Levenshtein distance) between the two strings divided by the length of the longer string. This normalization ensures that the metric is scale-invariant and ranges between 0 and 1.

The RED is defined as:

$$\text{RED} = \frac{\text{EditDistance}(S_1, S_2)}{\max(|S_1|, |S_2|)},$$

where

- EditDistance$(S_1, S_2)$ is the Levenshtein distance between strings $S_1$ and $S_2$, which measures the minimum number of single-character edits (insertions, deletions, or substitutions) required to transform $S_1$ into $S_2$.
- $|S_1|$ and $|S_2|$ are the lengths of strings $S_1$ and $S_2$, respectively.
- $\max(|S_1|, |S_2|)$ is the length of the longer string, used to normalize the edit distance.

The RED score ranges from 0 to 1, where 0 indicates that the two strings are identical (no edits are needed) and 1 indicates that the two strings are completely dissimilar (every character needs to be edited).

### D.1.4. PASS@$k$ METRIC

The Pass@$k$ metric is a performance evaluation metric used to assess the quality of code generation models. It measures the probability that at least one correct solution is generated within the top $k$ samples produced by the model. This metric is particularly useful for evaluating models in scenarios where multiple candidate solutions are generated, and the goal is to determine how often the model produces a correct solution within a limited number of attempts.

Given a set of $n$ generated samples for a problem, the Pass@$k$ metric is calculated as follows:

$$\text{Pass@}k = \frac{\text{Number of problems with at least one correct solution in the top } k \text{ samples}}{\text{Total number of problems}}.$$

Alternatively, if the model generates $k$ samples per problem, the Pass@$k$ metric can be computed as:

$$\text{Pass@}k = \mathbb{E}_{\text{problems}} \left[ 1 - \frac{\binom{n-c}{k}}{\binom{n}{k}} \right],$$

where

- $n$ is the total number of samples generated per problem.
- $c$ is the number of correct solutions among the $n$ samples.
- $\binom{n-c}{k}$ is the number of ways to choose $k$ samples that do not contain any correct solutions.
- $\binom{n}{k}$ is the total number of ways to choose $k$ samples from $n$.

The Pass@$k$ metric ranges from 0 to 1, where 1 indicates that at least one correct solution is always found within the top $k$ samples and 0 indicates that no correct solution is ever found within the top $k$ samples.

### D.2. RQ2. Experiment Settings

In the process of RQ2, we train five open-source models using five knowledge update techniques, and evaluate trained models on CODESYNCBENCH. In this section, we show the detailed experiment settings as follows.

### D.2.1. MODEL TRAINING

**Knowledge Update Methods.** Supervised Fine-Tuning (SFT) is a widely used and traditional method for modifying and aligning model knowledge, relying on labeled data to train models. For the SFT training dataset, the *context* in metadata serves as the prompt, and the *updated_data* serves as the target sequence. We also evaluate three instruction tuning methods (e.g., DPO (Rafailov et al., 2023), ORPO (Hong et al., 2024), SimPO (Hong et al., 2024)) to update the knowledge, relying on positive-negative data pairs to train models. For their training datasets, we use *updated_code* and *outdated_data* as the positive and negative target sequences respectively. We use LoRA for all instruction tuning experiments (**?**) based on LoRA SFT on A800 servers.

We adopt five knowledge update techniques: SFT, SFT (LoRA), DPO, ORPO, SimPO. Additionally, LoRA training requires less computation resources and is possessed of high efficiency.

We train DPO, ORPO and SimPO using LoRA techniques, which is more efficient than that of full training. We use LLaMA-Factory (Zheng et al., 2024b), a user-friendly and reliable automated tuning framework.

**Hyperparameter.**

Table 7: **RQ2. Hyperparameters for Qwen2.5-7B-Instruct**

| Techniques | Epoch | Learning Rate | Warmup Ratio | Preference Beta |
|---|---|---|---|---|
| SFT | 3 | 1.0e-4 | 0.1 | – |
| SFT(LoRA) | 3 | 1.0e-4 | 0.1 | – |
| DPO | 3.5 | 5.0e-6 | 0.1 | 0.1 |
| ORPO | 3.5 | 5.0e-6 | 0.1 | 0.1 |
| SimPO | 3.5 | 5.0e-6 | 0.1 | 0.1 |

### D.2.2. EVALUATION ON HUMANEVAL

We utilize the open-source project Code Generation LM Evaluation Harness (Ben Allal et al., 2022) to assess our models on the HumanEval benchmark (Chen et al., 2021b). This evaluation framework provides a standardized method for measuring the code generation capabilities of LLMs.

For each evaluation, we generate 10 independent samples per problem across all 164 programming tasks in the benchmark. We then compute the Pass@1, Pass@3, and Pass@5 metrics, which measure the probability of generating a correct solution within the top 1, 3, or 5 model outputs, respectively.

To further analyze model performance, we calculate the Pass@5 ratio between the trained models and the reference models. This comparison, visualized in Figure 2, serves as a diagnostic tool to monitor the effectiveness of our training experiments. The results indicate that all models perform on par with the reference models, suggesting that catastrophic forgetting is minimal. Moreover, our approach successfully injects new knowledge into the models without degrading their existing capabilities.

This evaluation provides strong evidence that our training strategy effectively balances knowledge retention and expansion, ensuring that models maintain their baseline performance while learning new information.

### D.3. RQ3-1. Experiment Settings

Retrieving invocation instances for each API presents challenges due to the limited number of available instances, which complicates the scaling of both training sets and benchmarks. In most cases, we only have access to a small number of instances. On the other hand, a limited sample size may lead to underfitting, while a larger sample size does not necessarily equate to better performance. In this section, we evaluate the impact of sample size on model performance.

To address this, we prepare a series of training sets, each containing the same APIs but varying numbers of samples per API. Specifically, we explore four different sample sizes: 5, 10, 20, and 50, representing different levels—low, medium, high, and very high.

We construct these training sets from the original dataset. To control the experimental conditions, all four sets are derived from the same set of APIs. Consequently, we include APIs that have more than 50 samples. We then randomly select a fixed number of samples for each API. To reduce sample quality variance, we ensure that the sets overlap. For example, the 5-sample set is fully included in the 10-sample set, and so on.

Next, we train the model Qwen2.5-7B-Instruct (Qwen Team, 2024) on these sets. Due to the limited size of the subsets, we double the number of epochs (which was set to 3 in Appendix D.2, and thus set to 6 for this experiment). To ensure convergence of the loss value, we adjust the relevant hyperparameters, as shown in Table 8.

Table 8: **RQ3-1. Hyperparameters for Qwen2.5-7B-Instruct across different training datasets.**

| Counts | Technique | Eval Steps | Learning Rate | Preference Beta |
|--------|-----------|------------|---------------|-----------------|
| **5** | SFT(LoRA) | 30 | 1.0e-5 | – |
| | DPO | 30 | 5.0e-6 | 0.3 |
| | ORPO | 30 | 5.0e-6 | 0.1 |
| | SimPO | 30 | 5.0e-6 | 0.7 |
| **10** | SFT(LoRA) | 50 | 1.0e-5 | – |
| | DPO | 50 | 5.0e-6 | 0.3 |
| | ORPO | 50 | 5.0e-6 | 0.1 |
| | SimPO | 50 | 5.0e-6 | 0.7 |
| **20** | SFT(LoRA) | 200 | 1.0e-5 | – |
| | DPO | 200 | 5.0e-6 | 0.3 |
| | ORPO | 200 | 5.0e-6 | 0.1 |
| | SimPO | 200 | 5.0e-6 | 0.7 |
| **50** | SFT(LoRA) | 500 | 1.0e-5 | – |
| | DPO | 500 | 5.0e-6 | 0.3 |
| | ORPO | 500 | 5.0e-6 | 0.1 |
| | SimPO | 500 | 5.0e-6 | 0.7 |

## D.4. RQ3-2. Experiment Settings

LLMs demonstrate varying capabilities across different categories of APIs. To align with RQ2 (see Appendix D.2), we evaluate the trained models from RQ2 on different subsets of CCT within CODESYNCBENCH. Specifically, we categorize CCT in CODESYNCBENCH into three distinct groups based on API types: functions, methods, and initializers. Each trained model is assessed separately on these subsets to analyze its performance across different API structures.

To ensure a fair and robust evaluation, we set the temperature to 0.9 and generate five output samples per prompt to account for variability in model responses. The model outputs are then compared against reference answers using BLEU scores, which serve as a metric for measuring output accuracy. The results of this evaluation are presented in Figure 7, providing insights into how model performance varies across API categories.

This analysis helps us understand whether LLMs exhibit strengths or weaknesses in handling specific API types, offering valuable guidance for improving future models and fine-tuning strategies.

# E. Data Format

## E.1. MetaData Format

```
MetaData

[API] torch.optim.swa_utils.AveragedModel.load_state_dict

[Code Context]
def load_model_from_state_dict(state_dict, input_dim=None):
    model = optim.swa_utils.AveragedModel(SNN(input_dim=input_dim,
    num_hidden_units=hidden_dim))
    model.load_state_dict

[Updated Code]  (state_dict, strict=True, assign=False)
[Outdated Code] (state_dict, strict=True)
```

## E.2. Training Data Format

### E.2.1. SFT TRAINING DATA

```
SFT Training data
[instruction]
Please fill the parameter list of api
\"torch.optim.swa_utils.AveragedModel.load_state_dict\"
according to the given context.

[input]
def load_model_from_state_dict(state_dict, input_dim=None):
    model = optim.swa_utils.AveragedModel(SNN(input_dim=input_dim,
    num_hidden_units=hidden_dim))
    model.load_state_dict

[output] (state_dict, strict=True, assign=False)
```

### E.2.2. DPO/ORPO/SIMPO TRAINING DATA

```
DPO/ORPO/SimPO Training data
[conversations]
    [from] system
    [value] Please complete subsequent API calling statement.

    [from] human
    [value]
    def load_model_from_state_dict(state_dict, input_dim=None):
        model = optim.swa_utils.AveragedModel(SNN(input_dim=input_dim,
        num_hidden_units=hidden_dim))
        model.load_state_dict

[chosen]
    [from] gpt
    [value] (state_dict, strict=True, assign=False)

[rejected]
    [from] gpt
    [value] (state_dict, strict=True)
```

### E.3. Code Completion Task Format

```
[API_path] flask.json.dump
[question]
def test_json_dump_to_file(self):
    app = flask.Flask(__name__)
    test_data = {'name': 'Flask'}
    out = StringIO()
    with app.app_context():
        flask.json.dump
[answer] (test_data, out)
```

### E.4. Error Correct Task Format

```
[API_path] flask.json.dump
[question]
def test_json_dump_to_file(self):
    app = flask.Flask(__name__)
    test_data = {'name': 'Flask'}
    out = StringIO()
    with app.app_context():
        flask.json.dump(token_data, file, app=None)
[answer] (token_data, file)
```

## E.5. Multiple Choice Question Format

```
[API_path] flask.json.dump
[question]
def test_json_dump_to_file(self):
    app = flask.Flask(__name__)
    test_data = {'name': 'Flask'}
    out = StringIO()
    with app.app_context():
        flask.json.dump
[A] (test_data, out, app=app)
[B] (test_data, out)
[C] (test_data, out, app=app, indent=4)
[D] (test_data, out, app=None)
[answer] B
```

