# OpenReview forum: "CodeSync: Synchronizing Large Language Models with Dynamic Code Evolution at Scale"
_ICML.cc/2025/Conference — ICML 2025 poster_

### Official Review · Reviewer_Jxkj · 2025-03-09

**Overall Recommendation:** 4

**Summary:**

This paper introduces CodeSync, a data engine for identifying outdated code patterns and collecting real-time code knowledge updates from Python third-party libraries. Building upon CodeSync, the authors further develop CodeSyncBench, a comprehensive benchmark for assessing LLMs’ ability to stay synchronized with code evolution, which covers real-world updates for 220 APIs from six Python libraries. Extensive experiments on 14 state-of-the-art LLMs reveal that they struggle with dynamic code evolution, even with the support of advanced knowledge updating methods (e.g., DPO, ORPO, and SimPO). This benchmark offers a strong foundation for the development of more effective methods for real-time code knowledge updating in the future.

**Claims And Evidence:**

The paper introduces an interesting benchmark to evaluate the synchronized ability of LLMs for dynamic code evolution. The experiments conducted in Section 2.4 provide convincing evidence to support the proposal.

**Essential References Not Discussed:**

The related works are sufficiently discussed in this paper.

**Experimental Designs Or Analyses:**

I have checked the experimental designs and analysis

**Methods And Evaluation Criteria:**

The pipeline of constructing the dataset and the evaluation criteria of the benchmark seem reasonable.

**Other Comments Or Suggestions:**

No

**Other Strengths And Weaknesses:**

Pros:
1. This paper is well presented with a clear structure.
2. The studied problem of synchronizing LLMs with dynamic code evolution is well motivated and interesting.
3. The experiments conducted in this paper are comprehensive, and several interesting findings are delivered in this paper.

Cons:
1. This paper assumes that LLMs should always be updated with the latest API versions. However, it may not align with real-world development needs, since many projects continue using order versions for stability and compatibility. If CodeSync only fine-tuned on the latest API changes, it might lead to over-specialization.
2. As LLMs generate legacy-updated API pairs, they may introduce errors. If the datasets include incorrect API updates, this could mislead LLM training and evaluation, compromising the reliability of the benchmark.
3. Library evolution is a continuous process that not only involves modifying existing APIs but also includes introducing new functionalities and deprecating outdated APIs. The paper discusses API changes but does not explain how CodeSync deals with APIs that are removed without direct replacements (deprecated APIs).

**Questions For Authors:**

1. The paper provides limited details on how 220 APIs are selected from the 6,036 tracked updates. What specific criteria are used for filtering? More details on building the dataset would be beneficial to assess the benchmark's representativeness and reliability.

2. Since libraries continue to evolve, how will CodeSyncBench remain up to date? Is there a plan for periodic updates, or will the dataset become outdated as newer API changes emerge?

3. What is the "Parameter Mapping" that is referred to in Appendix B.1.2. Would this method include some APIs that are not updated?

4. Could the authors provide a clear and detailed explanation of the process for identifying valid method API invocations, given that method APIs often involve dynamic binding?

**Relation To Broader Scientific Literature:**

The presented novel dataset and benchmark for dynamic code evolution can serve as a complement to the existing progress on LLMs for code intelligence.

**Theoretical Claims:**

No

---

> ### Author Rebuttal · Authors · 2025-03-31
>
> **Dear Reviewer Jxkj,**
>
> We would like to express our sincere gratitude for your thoughtful and constructive feedback. We have addressed all of the comments and thoroughly presented our most recent experimental findings.
>
> ---
> **Q1:** The paper provides limited details on how 220 APIs are selected from the 6,036 tracked updates. What specific criteria are used for filtering? More details on building the dataset would be beneficial to assess the benchmark's representativeness and reliability.
>
> **A1:** As detailed in Appendix B.1, we filtered out APIs with insufficient real-world usage. Specifically, we selected 220 APIs that each had at least 15 retrieved real-world invocation examples.
>
> ---
> **Q2:** Since libraries continue to evolve, how will CodeSyncBench remain up to date? Is there a plan for periodic updates, or will the dataset become outdated as newer API changes emerge?
>
> **A2:** Yes, we plan to maintain and periodically update CodeSyncBench to reflect ongoing API changes, ensuring it remains a relevant and reliable benchmark for evaluating LLMs' ability to stay synchronized with evolving API knowledge over time.
>
> ---
> **Q3:** What is the "Parameter Mapping" that is referred to in Appendix B.1.2. Would this method include some APIs that are not updated?
>
> **A3:** As described in Appendix B.1.2, parameter mapping is a standard technique in software engineering for detecting API signature changes. For each pair of APIs across versions, we analyze:
> - Parameter types: positional-only, keyword-only, or both
> - Parameter attributes: required vs. optional
>
> We then construct a mapping from old-version parameters to their corresponding new-version parameters. If no mapping can be established—for example, due to newly added or removed parameters—this indicates an update. Additionally, if mapped parameters differ in type or attribute, we classify the API as changed. We will clarify this in the next version of our manuscript.
>
> ---
> **Q4:** Could the authors provide a clear and detailed explanation of the process for identifying valid method API invocations, given that method APIs often involve dynamic binding?
>
> **A4:** As stated in Appendix B.1, we initially collected 6,036 updated APIs and retrieved invocation samples for each. The next step is to identify target API invocations, as data retrieval relies on string match. We then analyze the code context to determine the actual API calls. For Method-Level APIs, precisely identifying their invocations through static analysis alone is extremely challenging. Therefore, we limit our focus on cases where parameter types can be inferred from the code context. Specifically, we track the dataflow and control flow of each instance. Then, we bind class type and identify method calls for each. This method makes sure to capture actual API calls.
>
> ---
> **W5:** This paper assumes that LLMs should always be updated with the latest API versions. However, it may not align with real-world development needs, since many projects continue using older versions for stability and compatibility. If CodeSync only fine-tuned on the latest API changes, it might lead to over-specialization.
>
> **A5:** Thanks for your comments. We think there may be a misunderstanding here. Actually, *CodeSyncBench* selected the current time as a snapshot to highlight the importance of API evolution. However, our framework is highly flexible—users can specify any preferred version ranges of a library to generate customized benchmarks. As we have shown below, users can set different start times and deadlines of API evolution for each library. The training set generated by CodeSync helps LLMs to synchronize with target API knowledge. Therefore, in the real-world development scenarios, users can design their own training sets and benchmarks and fine-tune models using LoRA methods for further application and exploration.

---

### Official Review · Reviewer_oxwH · 2025-03-16

**Overall Recommendation:** 2

**Summary:**

The paper proposes a new benchmark called CODESYNC to address the issue that in the real world, library functions evolve over time while LLM code generation models are not updated. They source the API function updates from 6 real-world repositories: pandas, numpy, scipy, tensorflow, torch, and flask. They collect the API function updates by comparing the function signatures between the latest version and the version from Jan 2023, which is the ChatGPT release date. They also source the invocations of these functions from GitHub and use LLMs to create old-version invocation code from the new-version invocation code. The resulting 1100 legacy-updated pairs are used to create 1100 code completion, error correction, and multiple-choice questions. The other 2200 pairs are used as training data for updating LLMs' library function knowledge. The results are evaluated across many open and closed LLMs. They also finetune open code LLMs with SFT, DPO, ORPO, SimPO, etc. to see if the finetuned models can improve performance on the CODESYNC benchmark.

**Claims And Evidence:**

- The paper claims to propose a data engine that systematically collects real-time API updates from various Python third-party libraries, and they show that they can use the method to collect many real-world API updates.
- The paper claims to propose a novel benchmark for evaluating the performance of LLM code generation models, which is indeed different from previous API-update benchmarks like CodeUpdateArena.
- The paper claims to have comprehensive evaluation, and they did show model evaluations across many open and closed LLMs.

**Essential References Not Discussed:**

There are many many knowledge editing work in NLP which is not discussed.

**Experimental Designs Or Analyses:**

The design of the benchmark heavily sources from existing real-world repos instead of relying too much on synthetic data like the previous related work such as CodeUpdateArena, which makes it valuable to measure the real-world API update impact on LLM code generation. However, the paper only focuses on function signature changes, whereas API changes may also include changes to function semantics. Another limitation of the benchmark is the fixed legacy version (Jan 2023), which may favor LLMs with later knowledge cutoff dates. Finally, the paper only includes 6 popular Python libraries, which may be a limitation of the benchmark.

**Methods And Evaluation Criteria:**

The paper uses BLEU, ROUGE, and relative edit-distance scores for the code completion and error correction tasks. Not including exact match score or semantic-aware matching score (exact match with respect to parameter ordering) is not ideal.

**Other Comments Or Suggestions:**

Please see the questions below.

**Other Strengths And Weaknesses:**

The use of real-world API changes and invocations makes this a very valuable benchmark for evaluating the performance of LLMs in the midst of real-world API changes. However, the paper only focuses on function signature changes, whereas API changes may also include changes to function semantics. Another limitation of the benchmark is the fixed legacy version (Jan 2023), which may favor LLMs with later knowledge cutoff dates. Please also see the questions below.

**Questions For Authors:**

- Why is the HumanEval score for Qwen2.5-Coder-7B-Instruct so low? The results from the Qwen team show that it achieves 88.4% on HumanEval.
- How is the training data related to the benchmark? Does every function change in the benchmark problem have a corresponding data point in the training set?
- Is it possible to include RAG + prompting baseline?
- How much human effort is needed for creating the synthetic legacy update code from the latest API invocation data?

**Relation To Broader Scientific Literature:**

The paper is related to previous work on API update benchmarks like CodeUpdateArena. It also relates to knowledge update work in the NLP literature.

**Theoretical Claims:**

N/A

---

> ### Author Rebuttal · Authors · 2025-03-31
>
> **Dear Reviewer oxwH,**
>
> We sincerely appreciate your thoughtful review and your recognition of our work’s contributions to the community.
>
> ---
> **Q1:** Why is the HumanEval score for Qwen2.5-Coder-7B-Instruct so low? The results from the Qwen team show that it achieves 88.4% on HumanEval.
>
> **A1:** In fact, the HumanEval score we reported for Qwen2.5-Coder was obtained using the `bigcode-evaluation-hardness` framework, which applies stricter evaluation criteria than those used by the Qwen team. Notably, it does not automatically add import statements and enforces stop tokens, leading to a more rigorous evaluation.
>
> To ensure fairness, we have updated our evaluation setup to align more closely with the official criteria and re-evaluated all base and updated models. The revised scores are as follows:
>
> |Model|Base|SFT|DPO|SimPO|ORPO|
> |--- |---|---|---|---|---|
> |Qwen|65.24|62.80|61.59|63.41|63.41|
> |Qwen-Coder|82.32|82.32|82.93|82.93|81.71|
> |Llama|62.20|60.98|58.54|62.20|60.37|
> |CodeLlama|38.41|36.59|36.59|35.98|35.37|
> |DeepSeek-Coder|72.56|71.34|70.12|68.29|68.29|
>
> The updated results align with trends from the original settings, indicating that our conclusions remain valid.
>
> ---
>
> **Q2:** How is the training data related to the benchmark? Does every function change in the benchmark problem have a corresponding data point in the training set?
>
> **A2:** As stated in lines 255–262, for each API update, we collected 15 legacy-updated invocation pairs—10 for training and 5 for evaluation—ensuring that every API change in the benchmark has corresponding data points in the training set. However, this does not imply data leakage, as the training and benchmark data points use distinct code contexts, as discussed in our response to **Reviewer HLiK (W4)**.
>
> ---
> **Q3:** Is it possible to include RAG + prompting baseline?
>
> **A3:** Thank you for the suggestion. We have included RAG baseline results in our response to **Reviewer HLiK (Q3)**.
>
> ---
> **Q4:** How much human effort is needed for creating the synthetic legacy update code from the latest API invocation data?
>
> **A4:** As stated in Sec. 2.3, to ensure benchmark quality, two authors manually verified the divergence between legacy and updated API invocations synthesized by DeepSeek-V3. On average, verifying and guiding re-synthesis required \~1 minute per invocation pair.
>
> ---
> **W5:** The paper uses BLEU, ROUGE, and relative edit-distance scores for the code completion and error correction tasks. Not including exact match score or semantic-aware matching score (exact match with respect to parameter ordering) is not ideal.
>
> **A5:** Thank you for the suggestion. We have addressed this concern by incorporating AST-based semantic matching in our evaluation in our response to **Reviewer HLiK (Q1)**.
>
> ---
> **W6:** The paper only focuses on function signature changes, whereas API changes may also include changes to function semantics.
>
> **A6:** While API updates that involve purely semantic changes without signature modifications do occur, they are relatively uncommon and undocumented. Thus, our benchmark mainly focuses on API signature changes, as these typically reflect substantial semantic modifications and can be systematically tracked.
>
> ---
> **W7:** Another limitation of the benchmark is the fixed legacy version (Jan 2023), which may favor LLMs with later knowledge cutoff dates.
>
> **A7:** While LLMs with later knowledge cutoffs may have greater exposure to recent API updates, this does not contradict the benchmark’s objective—evaluating LLMs' ability to access and adapt to real-time API updates. In fact, this setup reflects a realistic scenario for assessing up-to-date knowledge. Nevertheless, even models like DeepSeek-R1 with recent cutoffs (July 2024), achieve low BLEU scores (e.g., 19.32), indicating limited effectiveness in handling evolving APIs. This further supports the need for explicit knowledge updating methods.
>
> ---
> **W8:** The paper only includes 6 popular Python libraries, which may be a limitation of the benchmark.
>
> **A8:** Although CodeSyncBench currently focuses on six widely-used Python libraries, our proposed CodeSync engine is generalizable. It can track API updates across any Python library over arbitrary time periods. Thus, the benchmark can readily be extended by broadening the version range and including more libraries as needed. We leave the extension to our future work.

---

### Official Review · Reviewer_HLiK · 2025-03-19

**Overall Recommendation:** 2

**Summary:**

The paper introduces CodeSyncBench, a benchmark to evaluate LLMs’ abilities to invoke the most recent versions of Python APIs. The benchmark shows that LLMs struggle to invoke APIs in the benchmarks correctly as measured by the chosen metrics. Further, the authors use various alignment techniques such as SFT, DPO and SimPO to fine-tune the LLMs to invoke the APIs correctly.

**Claims And Evidence:**

The claims are more or less backed by concrete evaluations (see comments below). One concern that stands out for me is the lack of commentary on the leakage of samples from CodeSyncBench into the training data of models being evaluated.

**Essential References Not Discussed:**

The paper does not disregard contemporary works also focusing on API invocations [1, 2] and benchmarks developed [3, 4]. Further, the paper does not consider RAG, a popular and simple approach to updating API invocation knowledge [5], in their evaluations.

[1] Jain, Nihal, et al. "On Mitigating Code LLM Hallucinations with API Documentation." arXiv preprint arXiv:2407.09726 (2024).

[2] Patil, Shishir G., et al. "Gorilla: Large language model connected with massive apis." Advances in Neural Information Processing Systems 37 (2024): 126544-126565.

[3] Kuhar, Sachit, et al. "LibEvolutionEval: A benchmark and study for version-specific code generation." *arXiv preprint arXiv:2412.04478* (2024).

[4] Islah, Nizar, et al. "GitChameleon: Unmasking the Version-Switching Capabilities of Code Generation Models." *arXiv preprint arXiv:2411.05830* (2024).

[5] Asai, Akari, et al. "Self-rag: Learning to retrieve, generate, and critique through self-reflection." *The Twelfth International Conference on Learning Representations*. 2023.

**Experimental Designs Or Analyses:**

1. The evaluations conducted in the paper rely on alignment methods such as SFT, DPO, etc. While different methods improve performance over the baseline, the results among the methods appear largely mixed, and no concrete conclusion can be made about which method helps improve performance the most.
2. I believe that RAG must be considered as a baseline and is missing from the paper. If the experiment is out of scope, the paper should discuss when their method must be considered over RAG. It is now common knowledge to use RAG to update LLM knowledge and it’s unclear when the methods discussed in the paper would be adopted solely to update models’ knowledge about a few APIs.

**Methods And Evaluation Criteria:**

1. The metrics chosen for checking the validity of API invocations are dated and should no longer be used. It is well-known that these metrics (string match based criteria) suffer from not capturing the nuances of code syntax and semantics. I recommend the authors to instead use static analysis to verify the correctness of API invocations [1, 2, 3]. Measures like edit distance can be very high despite the model prediction being semantically equivalent to the ground truth.
2. There can typically be multiple ways of invoking the same API to achieve the same outcome, e.g., `np.sum([a, b]) <=> np.sum([b, a])` but the paper does not call these out or comment on how such edge cases are being handled during evaluations.

[1] Patil, Shishir G., et al. "Gorilla: Large language model connected with massive apis." Advances in Neural Information Processing Systems 37 (2024): 126544-126565.

[2] Ding, Yangruibo, et al. "Crosscodeeval: A diverse and multilingual benchmark for cross-file code completion." Advances in Neural Information Processing Systems 36 (2023): 46701-46723.

[3] Jain, Nihal, et al. "On Mitigating Code LLM Hallucinations with API Documentation." arXiv preprint arXiv:2407.09726 (2024).

**Other Comments Or Suggestions:**

Line 149 - thirt → third

**Other Strengths And Weaknesses:**

The text in Section 3 is largely a regurgitation of the results in the tables. The reader is not provided meaningful insights, if any, that can be drawn from the tables in the paper. Finally, the paper says that existing knowledge techniques “show limitations” without diving deep into these. It would be useful if the authors could comment on the “further refinements” that are needed. In fact, this should form the core focus of the paper as other details (such as SFT, DPO, etc. will improve performance somewhat) are well-known facts.

**Questions For Authors:**

1. How do the results of the paper change if you use static analysis methods, such as AST matching, etc. instead of string-matching approaches for evaluation? Incorporating these evaluations in your benchmarking package will improve trust in your benchmark.
2. Why do you think current knowledge updating methods discussed in the paper are limiting? What’s the expected upper-bound on performance using the current evaluations and metrics? What kind of improvements are you suggesting will improve performance further?
3. How does the simple RAG baseline compare with the methods discussed in the paper? When or when not should someone use your method as opposed to RAG to update API invocation knowledge?

**Relation To Broader Scientific Literature:**

This paper is related to generating updated API invocations with large language models. It provides a new benchmark to study this problem and surveys several post-training methods to address this problem. The paper’s findings that post-training methods slightly improve performance on the task is aligned with common knowledge about LLMs.

**Theoretical Claims:**

n/a

---

> ### Author Rebuttal · Authors · 2025-04-01
>
> **Dear Reviewer HLiK,**
>
> We sincerely appreciate your suggestions and assessment of our work! Motivated by your feedback, we are committed to improving our manuscript and providing a more comprehensive benchmark and evaluations!
>
> ---
> **Q1:** How do the results of the paper change if you use static analysis methods, such as AST matching, etc. instead of string-matching approaches for evaluation?
>
> **A1**: Thanks for the suggestion. Actually, we have considered CodeBLEU to measure the effectiveness of API invocation updating from the perspective of AST matching. However, we found that, in practice, many LLM completions contained invalid syntax or natural language, causing them to fail the AST parsing (e.g., \~70% of Qwen2.5-Coder completions on our benchmark failed AST parsing).
>
> As a result, we compute CodeBLEU using a fallback strategy: if an API invocation passes AST parsing, we compute CodeBLEU against the ground truth; otherwise, we assign a corresponding part score of zero. The results for Qwen2.5-Coder are shown below:
> |Method|CCT BLEU↑|CCT CodeBLEU↑|ECT BLEU↑|ECT CodeBLEU↑|
> |---|---|---|---|---|
> |Original|5.89|29.41|11.64|32.68|
> |SFT|15.44|31.17|19.20|36.03|
> |DPO|23.36|38.67|55.57|44.95|
> |ORPO|21.47|37.06|56.92|40.62|
> |SimPO|23.86|39.39|54.57|45.43|
>
> > The experimental results align with trends from the original metrics, indicating that our conclusions remain valid under structure-aware evaluation.  (Full results are available at: https://anonymous.4open.science/r/CodeSyncRebuttal-D65F ).
>
> ---
> **Q2:** Why do you think current methods are limiting? What’s the expected upper-bound on performance? What kind of improvements are you suggesting will improve performance further?
>
> **A2:** Thanks for your comments. Current knowledge-updating methods are limited in helping LLMs fully internalize and recall updated APIs, as evidenced by suboptimal CodeBLEU scores (<40) in the CCT task. The upper bound of performance can be quantified by evaluating LLMs’ ability to invoke legacy APIs. The empirical upper bounds for Qwen2.5 and Qwen2.5-Coder are as follows:
> |Model|Original|Best Updating Method|Upper Bound|
> |---|---|---|---|
> |Qwen|30.21|39.67|**42.05**|
> |QwenCoder|29.41|40.70|**45.12**|
>
> > The results indicate a significant performance gap with upper bounds. To close it, we recommend exploring hybrid approaches, as discussed in response to Q3.
>
> ---
> **Q3:** How does the simple RAG baseline compare with the methods discussed in the paper? When or when not should someone use your method as opposed to RAG to update API invocation knowledge?
>
> **A3:** As stated in lines 104–109, unlike RAG which increases inference overhead by retrieval for each query, our CodeSyncBench focuses on evaluating and improving LLMs’ ability to internalize API updates. FT methods allows models to update without extra costs while inference, making them preferable in latency-sensitive or offline scenarios. Moreover, RAG's reliance on retrieval quality is problematic, especially with large real-world API updates. In contrast, CodeSync tracks API changes and integrates them into LLMs, enabling more reliable recall of updated API knowledge.
>
> To address your concern, we test a RAG baseline with a vector database of 5,056 updated API signatures from six libraries mentioned in the paper, indexed with OpenAI’s `text-embedding-3-large` model.
>
> |Method|CCT CodeBLEU↑|ECT CodeBLEU↑|MCQ P@1↑|
> |---|---|---|---|
> |Original|29.41|32.68|32.56|
> |SFT|31.17|36.03|35.16|
> |DPO|38.67|44.95|37.00|
> |RAG|35.17|42.26|34.26|
> |SFT+RAG|**40.70**|**51.35**|**36.89**|
>
> > Results show that RAG performs lower than FT methods (35.17), primarily due to a retrieval success rate of only \~60% (665/1100). However, RAG achieves better results (40.70), while combined with FT methods, highlighting the potential of hybrid approaches. In summary, knowledge updating methods outperform RAG in scenarios requiring efficient inference and precise API knowledge tracking.
>
> We will incorporate these results and analysis into the revised manuscript.
>
> ---
> **W4:** One concern that stands out for me is the lack of commentary on the leakage of samples from CodeSyncBench into the training data of models being evaluated.
>
> **A4:** To demonstrate that there is no data leakage between training data and benchmark, we compute the n-gram similarity of all samples associated with each API. Lower n-gram (<15%) scores indicate minimal overlap and thus confirm the absence of leakage.
> |N-gram|N=5|N=10|N=15|N=20|
> |---|---|---|---|---|
> |Score|12.62|10.21|9.24|8.52|
>
> Specifically, after retrieving data for APIs, we duplicate all data and then split them into two parts. This approach has a clean separation and prevents any potential data leakage.
>
> **W5:** The paper does not include contemporary works also focusing on API invocations and benchmarks developed.
>
> **A5:** Thanks for the suggestion. We will include all the mentioned works in the next version of the paper.

---

### Decision · Program_Chairs · 2025-05-01

**Decision:**

Accept (poster)

**Comment:**

The reviewers agreed that this paper presents a useful dataset and positive results from its tuning experiments. They noted some concerns around the limited evaluation, in response to which the authors provided substantial expansions to the evaluations & results. This is appreciated and strengthens the work sufficiently to recommend its acceptance.